# Impact of smoke and non-smoke aerosols on radiation and low-level clouds over the Southeast Atlantic from co-located satellite observations

Alejandro Baró Pérez[1,2], Abhay Devasthale[3], Frida A.-M. Bender[1,2], and Annica M. L. Ekman[1,2]

[1]Department of Meteorology, Stockholm University, Stockholm, Sweden
[2]Bolin Centre for Climate Research, Stockholm, Sweden
[3]Atmospheric Remote Sensing, Research and development, Swedish Meteorological and Hydrological Institute, Norrköping Sweden

**Correspondence:** Alejandro Baró Pérez (alejandro.baro-perez@misu.su.se)

**Abstract.** Data derived from instruments onboard the Cloud-Aerosol Lidar and Infrared Pathfinder Satellite Observation (CALIPSO) and CloudSat satellites as well as meteorological parameters from reanalysis are used to explore situations when moist aerosol layers overlie stratocumulus clouds over the Southeast Atlantic during the biomass burning season (June to October). To separate and quantify the impacts of aerosol loading, aerosol type, and humidity on the radiative fluxes (including cloud top cooling), the data are split into different levels of aerosol and moisture loadings. The aerosol classification available from the CALIPSO products is used to compare and contrast situations with pristine air, with smoke, and with other (non-smoke) types of aerosols. A substantial number of cases with non-smoke aerosols above clouds are found to occur under similar meteorological conditions as the smoke cases. In contrast, the meteorology is substantially different for the pristine situations, making a direct comparison with the aerosol cases ambiguous. The moisture content is enhanced within the aerosol layers, but no monotonous increase of the relative humidity with increasing aerosol optical depth is found. Shortwave (SW) heating rates within the moist aerosol plumes increase with increasing aerosol loading and are higher in the smoke cases compared to the non-smoke cases. However, there is no clear correlation between moisture changes and SW absorption. Cloud top cooling rates do not show a clear correlation with moisture within the overlying aerosol layers due to the strong variability of the cooling rates caused by other meteorological factors (most notably cloud top temperature). No clear influence of aerosol type or loading on cloud top cooling rates is detected. Further, there is no correlation between aerosol loading and the thermodynamic structure of the atmosphere nor the cloud top height.

## 1 Introduction

Stratocumulus clouds have a cooling effect on Earth's climate due to their strong reflection of incoming solar radiation and their relatively small effect on the outgoing longwave radiation. The clouds tend to form under statically stable low tropospheric conditions and they are mainly maintained by longwave radiative cooling at the cloud top (Klein and Hartmann, 1993). The cloud top cooling creates turbulent overturning that mixes the boundary layer and allows the cloud to be fed by moisture from the surface. It also helps to preserve the temperature inversion immediately above the cloud top (Wood, 2012). Dark-coloured

aerosols, for example from biomass burning, efficiently absorb solar radiation (direct effect). This absorption alters the radiative fluxes and modifies the stability of the atmosphere, which in turn can affect cloud development and precipitation (semi-direct aerosol effect). Studies have shown that when absorbing aerosols are located above stratocumulus cloud decks, the shortwave heating of the aerosol layer tends to strengthen the inversion, which reduces the entrainment of dry air and leads to a moistened boundary layer with an increased liquid water content and more persistent clouds (Deaconu et al., 2019; Brioude et al., 2009; Johnson et al., 2004). On the contrary, if the absorbing aerosols are located within a cloud layer, they can reduce moisture and liquid water content via local shortwave heating, causing a reduction of the stratocumulus cloud cover (Deaconu et al., 2019; Hill et al., 2008). In addition to the direct and semi-direct effects, absorbing aerosols can act as cloud condensation nuclei (CCN) and affect the radiative properties and lifetime of the clouds (indirect effects, Twomey (1977); Albrecht (1989)). The overall climate impacts of the rich set of interactions between absorbing aerosols, clouds and radiation are not yet well understood and consequently not well represented by large-scale models (Deaconu et al., 2019). Model differences in aerosol and cloud properties lead to disagreeing forcing estimates, especially in regions where aerosols and clouds overlap (Zhang et al., 2019; Schulz et al., 2006).

From June to October, large amounts of biomass burning aerosols emitted by wildfires in the southwestern African Savanna are transported westwards over the Southeast Atlantic Ocean (De Graaf et al., 2020; Deaconu et al., 2019; Ichoku et al., 2003). The anticyclonic circulation typical of this region causes a broad area of subsidence over the cool waters of the upwelling zone in the ocean, producing one of the largest stratocumulus cloud decks on the planet (Formenti et al., 2019; Klein and Hartmann, 1993). Under usual conditions, the biomass burning aerosols are mostly advected over the marine boundary layer and hence above the stratocumulus clouds (Adebiyi et al., 2015). As these aerosols typically contain large amounts of soot (Chazette et al., 2019), the biomass burning season in the Southeast Atlantic offers an excellent opportunity to study the complex interactions between absorbing aerosols and clouds and to characterise their manifestations. Several studies have used satellite observations to investigate situations with absorbing aerosols above clouds. Some of them have analysed these situations on a global scale (e.g. Devasthale and Thomas (2011); Kacenelenbogen et al. (2019)) whereas others have focused on the Southeast Atlantic (e.g. Wilcox (2010); Costantino and Bréon (2013); Adebiyi et al. (2015); Deaconu et al. (2019)). The studies focused on the Southeast Atlantic have shown that an increase in the amount of absorbing aerosols above clouds results in a cloud fraction increase (Costantino and Bréon, 2013) and that the clouds are optically thicker in situations with high aerosol loadings (Deaconu et al., 2019). However, when using satellite observations, it is a complicated task to isolate the effects of aerosols on clouds from those caused by the background meteorology due to covariations between aerosols and meteorological conditions. The biomass burning aerosols are usually accompanied by an enhanced humidity associated with the outflow from the continental boundary layer (Haywood et al., 2004; Adebiyi et al., 2015; Zhou et al., 2017; Deaconu et al., 2019). The moisture, besides its potential impacts on aerosol ageing (Dubovik et al., 2002; Haywood et al., 2004; Kar et al., 2018; Deaconu et al., 2019), can also remotely affect the underlying clouds through the modification of radiative fluxes. For instance, large-eddy simulations and radiative transfer calculations have shown a reduction of the stratocumulus top longwave (LW) cooling due to a downward LW flux increase caused by the water vapour accompanying the aerosol layer (Yamaguchi et al. (2015), Zhou et al. (2017) and Deaconu et al. (2019)). This effect, combined with an increase of the atmospheric stability

due to shortwave (SW) absorption by the aerosols may decrease the entrainment rate (Deaconu et al., 2019), which impacts the deepening of the boundary layer and the transition from stratocumulus to cumulus (Wood, 2012).

In this work we use four years (2007-2010) of recently updated satellite datasets to further explore situations when moist aerosol layers overlie stratocumulus clouds over the Southeast Atlantic. We use retrievals derived from instruments onboard the Cloud-Aerosol Lidar and Infrared Pathfinder Satellite Observation (CALIPSO) and CloudSat satellites as well as meteorological parameters from the ERA5 reanalysis (Hersbach et al., 2020). We also use the CALIPSO aerosol discrimination algorithm to analyse the composition of the aerosol layers and to compare smoke versus non-smoke aerosol occurrences. One main goal of our study is to separate and quantify the impacts of aerosol loading, aerosol type, and humidity on the radiative fluxes within the aerosol layer as well as their potential influence on cloud top cooling. More specifically, we seek observational support for the model-based finding of reduced cloud top cooling from moist aerosol layers above the boundary layer. Furthermore, we examine if the loading and type of aerosol affect general cloud features such as cloud top height. In our study we use the satellite data products to select cases where aerosols and clouds are separated from each other. This was not explicitly done by Deaconu et al. (2019), who in their analysis included all aerosols above clouds occurrences close to the coast of Angola. Another feature of our study is that we explore in our data if the previously observed covariance between aerosol and moisture in the region implies a consistent and monotonous increase of humidity with aerosol loading. The observational data and methodology are described in Section 2. Our results are presented in Section 3 followed by a summary and conclusions in Section 4.

## 2    Datasets and methodology

### 2.1    CALIPSO, Cloudsat and ERA5

Table 1 displays a summary of the datasets, products and variables used in the study. The CALIOP (Cloud-Aerosol Lidar with Orthogonal Polarization) instrument on board CALIPSO provides information on aerosol and cloud optical properties with high vertical resolution. Furthermore, the CALIOP V4 classification algorithm (Kim et al., 2018), used in this work, discriminates between different types of aerosols and categorizes clouds as ice or water phase (Winker et al., 2009). The ice-water phase is derived from the volume depolarization ratio that allows to discriminate between spherical cloud droplets and non-spherical ice crystals (Winker et al., 2009). The aerosol type is determined using measurements of the integrated attenuated backscatter and the volume depolarization ratio as well as surface type and aerosol layer altitude and location (Omar et al., 2009). For each aerosol type, an extinction-to-backscatter ratio (lidar ratio) is determined based on measurements, modeling, and a cluster analysis of a multiyear Aerosol Robotic Network (AERONET) dataset (Omar et al., 2009; Kim et al., 2018). The aerosol lidar ratio allows calculation of the aerosol extinction from the lidar backscatter signals. The aerosol lidar ratio and aerosol classification were substantially improved in V4 of the algorithm compared to V3, which has contributed to reducing the aerosol optical depth differences between CALIOP and AERONET-MODIS (Moderate Resolution Imaging Spectroradiometer) ocean (Kim et al., 2018). The full set of tropospheric aerosol types identified by the algorithm (in V4) are: clean marine, dust, polluted continental/smoke, clean continental, polluted dust, elevated smoke and dusty marine. The lidar ratios used in the CALIOP V4 retrieval algorithm are identical for the "polluted continental/smoke" and the "elevated smoke" aerosols (70 sr at 532 nm and 30

sr at 1064 nm) (Kim et al., 2018). The only difference between the two aerosol types is the altitude of the aerosol layer (higher than 2.5 km for the elevated smoke and lower than the same altitude for polluted continental/smoke). The similarity between the smoke and the polluted continental aerosol types in the optical properties measured by CALIOP (depolarization and color ratio) makes these cases indistinguishable within the PBL. Thus, smoke aerosols can be present in both aerosol categories and
pollution lofted by convection or other mechanisms can be misclassified as "elevated smoke" (Kim et al., 2018).

Two datasets were used from the CALIPSO Version 4.20 (V4) Level 2 product: the Merged Aerosol and Cloud Layers Data and the Aerosol Profile Data. In the Merged Aerosol and Cloud Layers Data the information is reported by layers at a 5 km horizontal resolution. We used it in order to know the altitudes of the aerosol and cloud layers as well as the aerosol types. The Aerosol Profile Data provides information as profiles with 60 m and 5 km of vertical and horizontal resolution, respectively
and includes vertically resolved meteorological information derived from the Modern-Era Retrospective analysis for Research and Applications, Version 2 (MERRA-2). From this data set, we obtained the aerosol extinction and column optical depth of tropospheric aerosol (AOD in this study) at 532 nm. We additionally used the profiles of temperature, pressure and relative humidity ($RH$) for the specific analysis performed in section 3.6.

Previous studies have shown that the CALIOP V3 operational algorithm underestimates the optical depth at 532 nm of
aerosol layers above clouds compared to other sensors, particularly in the presence of thick aerosol layers (Jethva et al., 2014; Deaconu et al., 2017; Rajapakshe et al., 2017). Deaconu et al. (2017) found this underestimation to have a factor ranging from two to four depending on the aerosol type. In addition, Rajapakshe et al. (2017) found that the same algorithm probably overestimates the base of aerosol layers above clouds by 500 m. The strong attenuation of the backscatter signal at 532 nm caused by optically thick aerosol layers is the likely source of these biases: the problem can cause first an overestimation of
the aerosol layer bottom height, leading later to an underestimation of the optical thickness (Jethva et al., 2014; Deaconu et al., 2017; Rajapakshe et al., 2017). To our knowledge there is no detailed study regarding the uncertainty in CALIOP V4 retrievals of optical thickness and altitude of aerosols above clouds over the Southeast Atlantic. As precaution, we have in our study taken into account the bias found by Rajapakshe et al. (2017) related to the altitude of the base of the aerosol layers in V3 (see Section 2.3). There is no clear procedure for correcting the values of optical thickness. Thus we can only recognize that there
are uncertainties (pending to be studied in detail in the region) in the values of aerosol extinction/optical thickness used in our study. However, as we use V4, the retrievals of these variables should be improved compared to V3 (Kim et al., 2018).

The Cloud Profiling Radar (CPR) onboard CloudSat produces detailed images of cloud structures. The profiles of radiative fluxes and atmospheric heating rates used in our study were obtained from the 2B-FLXHR-LIDAR product (Henderson et al., 2013) which includes measurements from CALIPSO, Cloudsat and MODIS. In this product, the aerosol location and optical
depth are obtained from CALIPSO whereas the aerosol optical properties (including asymmetry parameter and single scattering albedo) are taken from D'Almeida et al. (1991) and a report by the World Meteorological Organization (WCP-55, 1983). Atmospheric state variables (surface pressure, surface temperature and profiles of pressure, temperature and specific humidity as well as ozone mixing ratio) needed by the 2B-FLXHR-LIDAR product are supplied by the CloudSat ECMWF-AUX data product. This product contains European Centre for Medium-Range Weather Forecasts (ECMWF) analyses data interpolated
to the Cloudsat CPR bins. The profiles of cloud ice and liquid water content are obtained from the CloudSat 2B-LWC and 2B-

**Table 1.** Satellite data and models used in the study. Variables with a star (*) are derived from the Modern-Era Retrospective analysis for Research and Applications, Version 2 (MERRA-2) data product. Variables with a two stars (**) are derived from the European Centre for Medium-Range Weather Forecasts (ECMWF) analysis.

| Satellite/reanalysis | Products and variables | Resolution |
|---|---|---|
| CALIPSO | **Merged Aerosol and Cloud Layers Data**: <br> -Aerosol top and base altitudes (km) <br> -Cloud top altitudes (km) <br> -Aerosol type. <br> **Aerosol Profile Data Products**: <br> -Extinction Coefficient at 532 nm <br> -Column Optical Depth Tropospheric Aerosols at 532 nm <br> -Temperature* <br> -Relative Humidity* <br> -Pressure* | Horizontal: 5 km <br> Vertical: 60 m |
| CloudSat | **2B-FLXHR-LIDAR Product**: <br> -Radiative fluxes <br> -Atmospheric heating rates <br> **ECMWF-AUX Product**: <br> -Temperature** <br> -Specific Humidity** <br> -Pressure** | Vertical: 240 m |
| ERA5 | wind speed and direction | Horizontal: 31 km |

IWC products, and the surface albedos are derived from seasonally-varying maps of surface reflectance properties. All this data is ingested into a radiative transfer model to compute the profiles of radiative fluxes at a vertical resolution of 240 m (Lebsock et al., 2017). From CloudSat we have also used the ECMWF-AUX data product (specifically the variables temperature, specific humidity and pressure) for the computation of the average profiles of potential temperature ($\theta$), specific humidity ($q_v$) and $RH$ in all the situations or cases analyzed in our study.

To carry out the analysis, the products obtained from the Merged Aerosol and Cloud Layers Data and the Aerosol Profile Data from CALIPSO were combined with the radiative fluxes, the atmospheric heating rates and the atmospheric state variables obtained from CloudSat. Since the spatial resolutions between the satellite data sets differ, the CloudSat profiles were averaged to the 5 km horizontal resolution of CALIPSO. Finally, the ERA5 reanalysis (Hersbach et al., 2020) was used to characterize the governing meteorological conditions during the period of analysis with special emphasis on winds.

## 2.2 Area and time period

The Southeast Atlantic area selected for the study extends from 10 to 18°S and from 2 to 10°E. It is located over the Namibian stratus region identified by Klein and Hartmann (1993) and is close to the continent, where the biomass burning aerosol loadings are high and where the aerosol layer is on average centered above the low-level clouds (Deaconu et al., 2019). The final extent of the area of study was determined based on a balance between having a sufficient number of cases while keeping the natural variability of meteorology and cloud properties relatively small. Our area of study is similar to the one used by Deaconu et al. (2019), but it is shifted 4° towards the west so that the entire domain is over the ocean. It is also 3° longer in the north-south direction.

The time period selected for the study is June to October for the years 2007 to 2010, i.e. covering the July-October period when the dominant winds frequently transport biomass burning aerosols from continental sources towards the stratocumulus decks located over the Southeast Atlantic (Adebiyi et al., 2015). Following Deaconu et al. (2019), who studied June to August (JJA) of one year (2008), we also included the month of June. Here, we divide the full biomass burning season into two parts, comparing the JJA period studied by Deaconu et al. (2019) with the September-October (SO) period, as a means of investigating how differences in meteorological conditions impact the manifestation of aerosol-cloud interactions.

## 2.3 Selection and classification of cases

To study the effects of aerosols overlying clouds, we identify and contrast cases with and without aerosols above clouds. We also distinguish between cases with smoke aerosols and aerosols with other optical properties using the CALIPSO V4 Level 2 product on aerosol and cloud layers (cf. Section 2.1) as follows:

1. **Smoke cases**: Atmospheric columns in which aerosol layers(s) classified by the CALIOP V4 algorithm as "elevated smoke" are above and detached from clouds. The main characteristics of these cases are:

   – The presence of only one cloud layer in the atmospheric column with cloud top altitude between 0.75km and 2.5km. Cases with cloud top altitudes lower than 0.75km are not considered to avoid the ground cluttered data in CloudSat retrievals. The maximum altitude (2.5km) was chosen to only capture scenarios with shallow clouds.

   – The presence of one or more aerosol layers above the cloud layer with a separation between the cloud layer and the bottom aerosol layer between 0.75 and 6 km. With the lower distance we expect to reduce the number of situations with possible contact between aerosols and clouds. This is the same distance used by Costantino and Bréon (2013) in their "well separated cases" (aerosol layers separated from cloud layers) and higher than the bias observed by Rajapakshe et al. (2017) for the altitudes of the bottom of aerosol layers above clouds. The higher altitude was selected to discard situations in which aerosols are very far from the clouds. Situations with more than one aerosol layer above cloud are included only if the distance between the aerosol layers is smaller than 0.3 km.

2. **Non-smoke cases**: Cases with aerosol layer(s) above clouds that are *not* categorized as "elevated smoke" by the CALIOP V4 algorithm. Otherwise, the same criteria as for the smoke category are used for the selection of the altitudes and number of aerosol and cloud layers.

3. **Pristine cases**: Cases containing only a cloud layer with a cloud top altitude between 0.75km and 2.5km, i.e. the same characteristics as described for the smoke cases (above) but with no aerosol present above the cloud layer.

## 3 Results

In this section, we will first examine the composition of the aerosol layer as determined by the CALIOP V4 retrieval algorithm. Next, we will examine if and how the spatial and temporal distribution of the aerosol and cloud layers differ between the three groups of cases (as defined in Section 2.3) and to what extent differences in the prevailing meteorological conditions may prevent a fair comparison between them. Thereafter, we will analyze the influence of the aerosol layer and its composition on the radiative heating profiles and examine the main drivers of any influence: aerosol type, loading or moisture ($RH$, $q_v$). Finally, selecting a greater number of aerosol situations (using less restrictions than those employed in 2.3), we will analyse the relationship between the aerosol optical depth and the free trophosperic moisture observed in our data and compare it with previous studies.

### 3.1 Aerosol type occurrence

The frequency of occurrence for the different aerosol types found within the aerosol layers above clouds are shown in figure 1. The "elevated smoke", which corresponds to the smoke cases in our study, is the predominant type, representing 56% and 61% of the total aerosol layers found during JJA and SO, respectively. Here we stress that it is possible to have polluted continental aerosol cases missclassified as "elevated smoke"(cf. section 2.3). We did not find, however, aerosol classified as "polluted continental". Among the remaining aerosol types (which correspond to the non-smoke cases in our study), the "Polluted dust" is predominant. The number of cases classified as "Elevated smoke" and "Polluted dust" is greater during SO than JJA. This happens because there is a maximum in the extent of the stratocumulus deck during September at the same time as there is a maximum in transport of continental aerosol over the Southeast Atlantic due to a strengthening of the anticyclone over southern Africa (Adebiyi et al., 2015).

Figure 1 shows that during the biomass burning season there is a non-negligible number of non-smoke aerosol cases overlying the stratocumulus clouds over the Southeast Atlantic. There is a possibility that some of these cases are misclassified by the CALIOP algorithm under certain circumstances (Kim et al., 2018). However, the aerosol classification has been improved from version V3 to V4 (used here), resulting in an increase of the aerosol classified as smoke over the Southeast Atlantic (Kar et al., 2018).

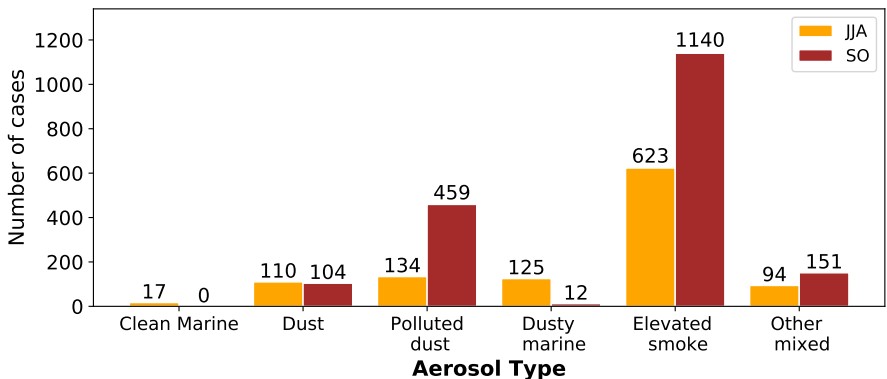

**Figure 1.** Number of profiles for each aerosol type found in the aerosol layers fulfilling the selection criteria (section 2.3) during the months June-July-August (JJA) and September-October (SO) during the period 2007-2010. All aerosol types, except "Other mixed", have the original name given by the CALIOP V4 algorithm. "Other mixed" refers to situations with more than one aerosol layer, where at least one of the layers is not defined as "elevated smoke". Our "smoke" cases correspond to "elevated smoke" whereas our "non-smoke cases" contain the rest of the aerosol types.

**Table 2.** Number of days and profiles used in the analysis for each case. The number of profiles in the smoke cases correspond to the "elevated smoke" shown in Figure 1. In the the non-smoke cases the number of profiles corresponds to the sum of the remaining aerosol types in Figure 1. Details on the definition of the cases are found in section 2.3.

| Periods analysed during years 2007-2010 | Number of days (number of profiles) analysed. | | |
|---|---|---|---|
| | Smoke | Non-smoke | Pristine |
| June-July-August (JJA) | 30 ( 623) | 31 (480) | 33 (705) |
| September-October (SO) | 42 (1140) | 43 (726) | 8 (218) |

## 3.2 Temporal and spatial distribution of cases

Next, we examine the number of cases identified and their spatial (horizontal and vertical) distributions during the two periods (JJA and SO). If these characteristics differ substantially, then the cases may also be subjected to different meteorological conditions which may influence the outcome of any comparison. Table 2 shows the total number of days and the total number of profiles when "smoke", "non-smoke" and "pristine" cases were found. The number of aerosol profiles is greater during SO than during JJA for the reasons explained in section 3.1. In contrast, pristine profiles are more frequent during JJA than SO.

The longitudinal and latitudinal distributions of all profiles are shown in Figure 2. Please note that the distributions are strongly influenced by our selection criteria (Section 2.3) and that we do not expect a complete agreement with a more general climatology of situations with aerosols above clouds or the cloud frequency distribution for the same region (e.g. Figures 3 and 4 in Devasthale and Thomas (2011)). During JJA, the aerosol cases are more numerous in the western (Figure 2a) and the northern parts of the area (Figure 2b), whereas the pristine cases are highest between 2-3 °E (i.e. the area farthest away from

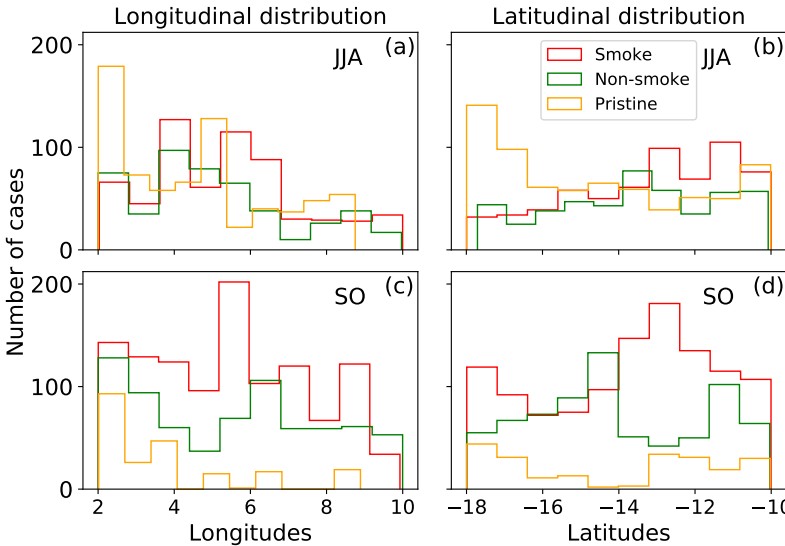

**Figure 2.** Longitudinal distributions (spanning latitudes from 10 to 18°S) and latitudinal distributions (spanning longitudes from 2 to 10°E) of the cases analysed during June-July-August (JJA)(a-b) and September-October (SO)(c-d) during the period 2007-2010.

the continent, Figure 2a) and south of 16 °S (Figure 2b). In SO all the cases are again more numerous in the western half of the area (Figure 2c). However, there are less similarities in their latitudinal distributions (Figure 2d).

The altitudes of the tops of the cloud layers are shown in Figure 3 together with the top and base altitudes of the aerosol layers. The average altitude of the cloud tops is clearly higher in the pristine cases (between 1.2 and 1.5 km) compared to both aerosol cases (around 1 km). For the aerosol cases the maximum cloud top altitude is close but below 1.5 km, a result consistent with Wilcox (2010). In contrast, the maximum cloud top altitude for the pristine cases is close to 2.2 km. Another notable feature is that the aerosol layer altitudes are on average higher during SO (4.2 km for smoke and 4.0 km for non-smoke cases) than during JJA (3.4 km for smoke and 3.1 km for non-smoke cases). Deaconu et al. (2019) obtained a similar result when comparing the periods May-July and August-October (with the later period having higher aerosol layer altitudes) for the years 2006 to 2010. We also note that all smoke cases have aerosol top altitudes higher than 2.5 km in accordance with the characteristics of the CALIOP V4 aerosol type "elevated smoke".

A likely cause of the difference in aerosol altitudes between JJA and SO is the location of the Southern African Easterly Jet. This jet supports biomass burning aerosol transport from the continent to the ocean and is stronger and migrates to higher altitudes (between 650 and 600 hPa) during SO (Adebiyi and Zuidema, 2016). Another factor that could contribute to the observed differences in the location of the aerosol layer is that land surface temperatures are higher in October (southern hemispheric spring) than in June (winter). Consequently, the top of the boundary layer, and the injection heights, may also be higher.

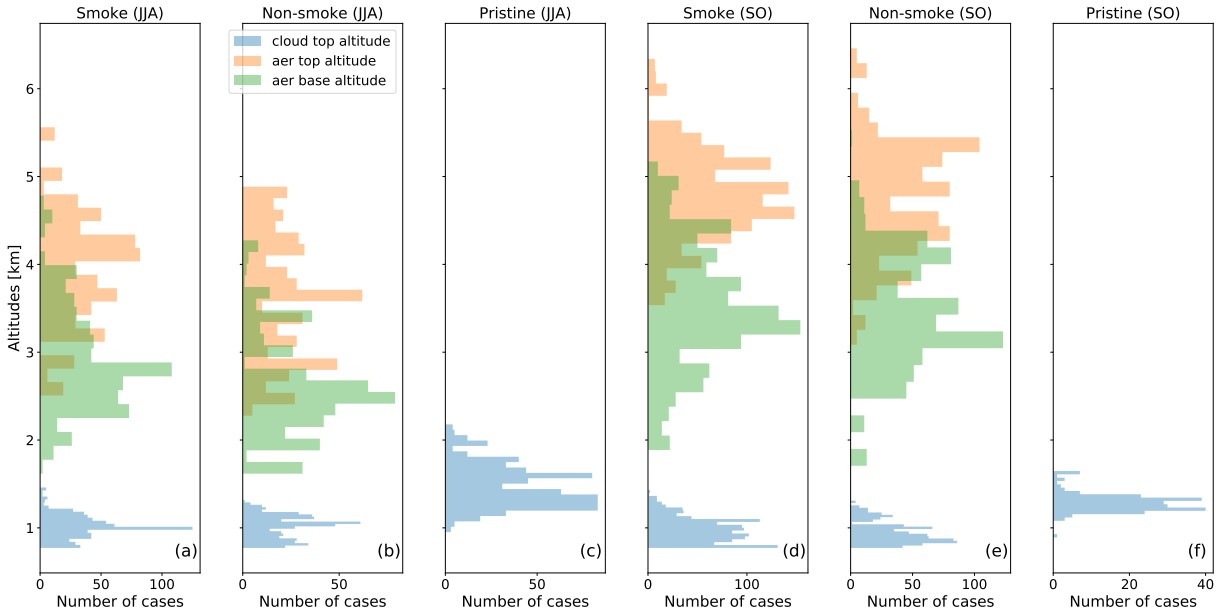

**Figure 3.** Altitudes of the cloud top and the aerosol (aer) top and base layers found within the area of study (latitudes from 10 to 18°S and longitudes from 2 to 10°E) during June-July-August (JJA) and September-October (SO) in the period 2007-2010. Aerosol cases are subdivided into smoke and non-smoke using the CALIOP V4 discrimination algorithm for the aerosol type.

### 3.3 Prevailing meteorological conditions

The atmospheric circulation governs the thermodynamic environment where clouds form. Even a small perturbation in the prevailing wind pattern may affect the temperature and humidity profiles and thereby the characteristics of a stratocumulus cloud layer (Wood, 2012). It is therefore important to ensure that the different groups of cases are subjected to similar large-scale circulation patterns and meteorology when investigating any influence of aerosol layers on the radiative fluxes and low-level cloud properties.

Figure 4 shows the average horizontal wind direction for the days corresponding to each of the three groups of cases at a level representative of the cloud layer (900 hPa during both JJA and SO) and a level representative of the aerosol layer (700 and 625 hPa during JJA and SO, respectively, cf. Figure 3). There are days containing both aerosol and pristine cases (i.e. one part of the satellite track within the area of study contains aerosol layer(s) above clouds whereas another part only contains clouds). The wind pattern on these days contributes to the horizontal wind average for all of the three cases (i.e. smoke, non-smoke and pristine). Figure A1 in the supplementary material shows a subset of the pristine cases (pristine*). These cases refers to days when pristine profiles were observed, but no aerosol above cloud case was detected at all along any satellite track within the area of study. This is an attempt to look at the wind patterns of completely pristine situations, although the possibility always exists of having aerosols above clouds in a part of the area of study not covered by the satellite trajectory.

In figure 4, the smoke and non-smoke cases have almost identical wind patterns, which is expected since they were often detected during the same days (not shown) and since their vertical distributions for both periods analysed were found to be similar (Section 3.2). At 900 hPa, southeasterly winds dominate during both JJA and SO. At 700 (625) hPa, the anticyclonic circulation imposes winds from the northeast (east) in JJA (SO), which favours transport of continental aerosol over the domain. In the pristine and pristine* cases winds are similar to the aerosol cases at 900hPa. However, at 700 and 625 hPa the influence of wind blowing from the open ocean becomes more pronounced (this is clearer in the pristine* than in the pristine cases), which prevents, or at least reduces, advection of aerosols over the area of study. Based on the analysis of the large-scale wind patterns, we draw the conclusion that the smoke and non-smoke aerosol cases experience similar large-scale circulation conditions while the pristine (and even more markedly the pristine*) cases do not, in particular in the free troposphere (625 and 700 hPa). Figure 4, together with a closer look at the wind speeds (not shown), also confirms that winds are stronger during SO compared to JJA, which is in agreement with the strengthening of the land-based anticyclone during SO and the strengthening and migration to higher altitudes of the Southern African Easterly Jet (Adebiyi et al., 2015; Adebiyi and Zuidema, 2016).

Figure 5a-d displays the average profiles of aerosol extinction, $RH$, $q_v$ and temperature for the different cases. The extinction is higher for the smoke than for the non-smoke cases. At aerosol layer altitudes, the northeasterly-easterly winds observed in figure 4 bring additional moisture together with aerosol resulting in higher $RH$ and $q_v$ values in the presence of aerosols above clouds compared to the pristine cases. This confirms previous studies associating the presence of aerosols above clouds with high moisture at aerosol layer altitudes in the region (e.g. Adebiyi et al. (2015); Deaconu et al. (2019)). Higher $RH$ and $q_v$ values are also observed in SO compared to JJA which can be linked to the strengthening of the easterlies during SO. During JJA, the $RH$ ($q_v$) within the aerosol layer is up to 8.4% (0.7 g/kg) higher for the smoke than for the non-smoke cases. The maximum difference in $RH$ ($q_v$) between the aerosol cases reduces to only 2.7% (0.3 g/kg) during SO. Even though $RH$ and $q_v$ differences are small, extinction differences reach 0.06 which is above the peak average extinction of the non-smoke cases (0.05) in JJA. The potential temperature profiles show a shallower boundary layer with a stronger inversion in the presence of aerosols compared to the pristine situations which supports the cloud top height differences observed in figure 3. It is likely that the difference in boundary layer height and cloud top altitudes is mainly caused by the northward shift of the anticyclonic circulation for the pristine cases (Figure 4).

In summary, while the two aerosol cases have similar meteorological conditions, the pristine (and pristine*) cases are less similar in terms of winds at 700 and 625 hPa (for JJA and SO respectively) and clearly differ in $RH$, $q_v$, and temperature profiles. These differences hamper the detection of any aerosol influence on cloud properties and radiative fluxes when comparing aerosol versus pristine cases.

## 3.4 Radiative heating profiles

Figure 5e-f shows the average radiative heating profiles for the different cases. The main difference between the smoke and non-smoke cases in net radiation is found within the aerosol layer, where the smoke cases show a clear average heating during both JJA and SO while the non-smoke cases only show an average heating during SO. The differences in net heating are mainly caused by a difference in the SW fluxes as the differences in the LW fluxes are small. Within the cloud layer, the net radiative

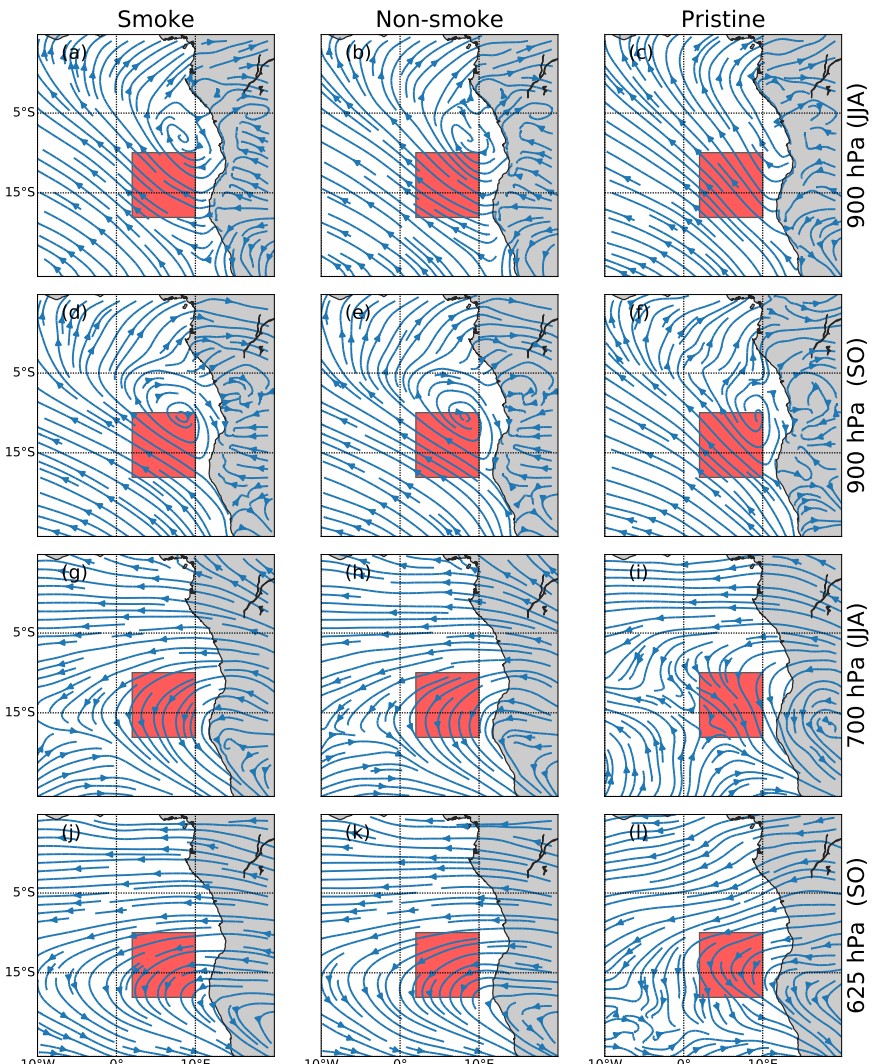

**Figure 4.** Streamlines corresponding to the average horizontal wind speed at 900 hPa (representative cloud level) in JJA and SO, and 700 (625) hPa (representative aerosol layer level) in JJA (SO) during the days when the smoke, non-smoke aerosol and pristine cases were identified during the period 2007-2010. The red square corresponds to the area of study.

heating for the pristine cases is, compared to the aerosol cases, higher during JJA and similar in magnitude during SO. The SW heating is higher in both seasons at the cloud layer of the pristine cases, whereas the LW cooling have a similar magnitude in JJA and higher values (more cooling) compared to the aerosol cases during SO. Above the boundary layer, the SW heating and the LW cooling are smaller for the pristine than for the aerosol cases; the net radiative heating is always negative in the pristine cases (above the boundary layer).

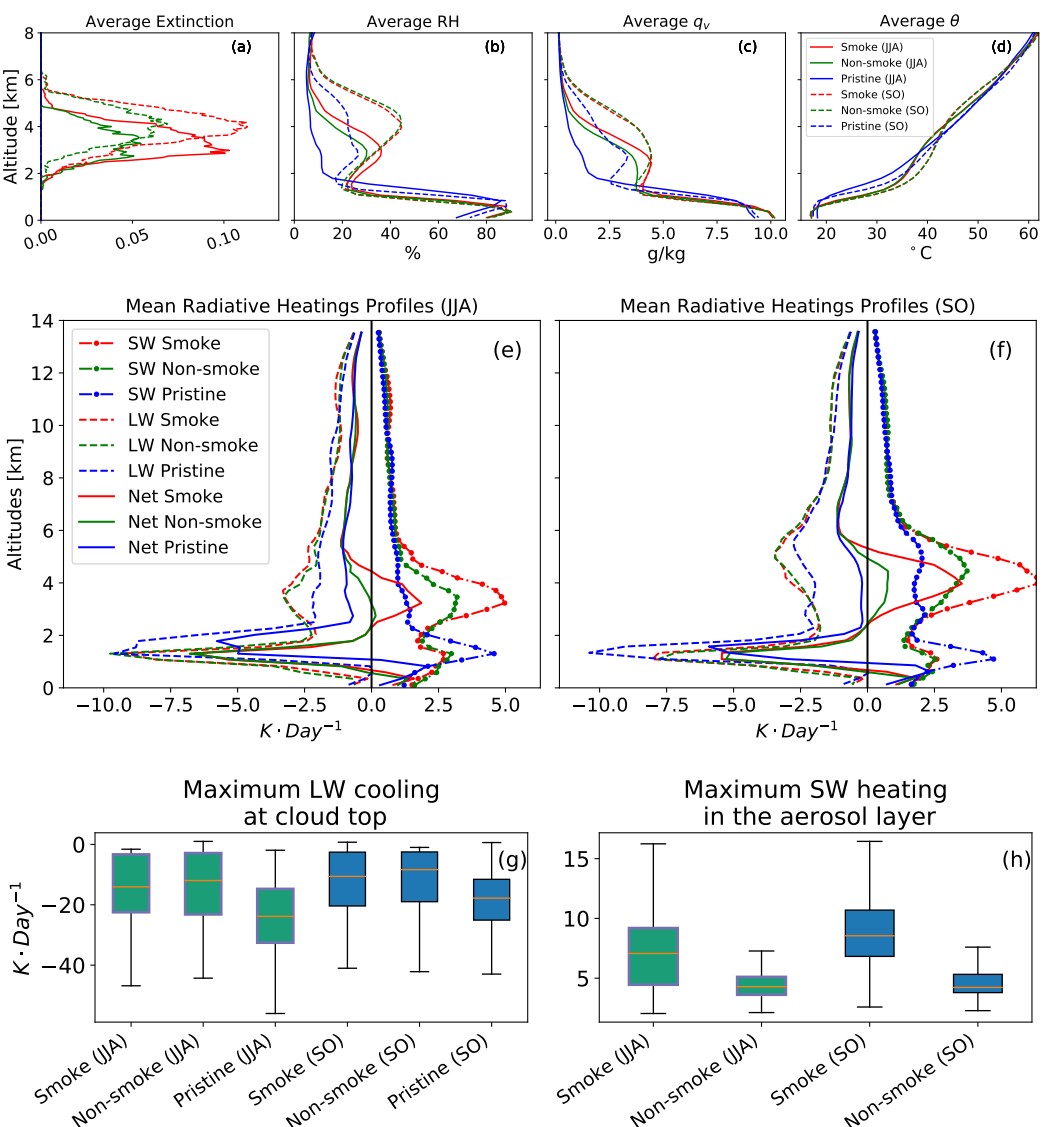

**Figure 5.** (a-d) Mean profiles of aerosol extinction, relative humidity ($RH$), specific humidity ($q_v$) and potential temperature ($\theta$) for the smoke, non-smoke and pristine cases during June-July-August (JJA) and September-October (SO) during the period 2007-2010. $RH$, $q_v$ and $\theta$ are derived from atmospheric variables in the CloudSat ECMWF-AUX product. (e-f) Mean shortwave (SW), longwave (LW) and net (Net) radiative heating profiles for the same cases and periods. (g-h) Box plots showing maximum, minimum, median, first and third quartiles of the maximum LW cooling at cloud top (g) and the maximum SW heating in the aerosol layer (h) for the smoke, non-smoke and pristine cases.

The average heating rates are sensitive to variations in the altitudes of the individual aerosol and cloud layers. We therefore identify the maximum LW cooling in the cloud layer (as a proxy for cloud top cooling) and the maximum SW heating in

the aerosol layer for each profile in each case and compare their distributions as box plots (figure 5g-h). No clear difference
is observed in the cloud top cooling rates between the aerosol cases, while the values of the median, first and third quartiles
are substantially lower for the pristine cases. A likely reason for the difference between the aerosol and pristine cases is the
difference in cloud top heights (cf. Figure 3 and discussion in Section 3.5). The mean maximum SW heating within the aerosol
layer is on average higher in the smoke cases than in the non-smoke cases. The spread in the heating rates is also larger which
is consistent with the wider aerosol optical depth (AOD) range observed in the smoke cases compared to the non-smoke cases
(not shown).

## 3.5   Influence of aerosol loading, moisture and cloud top altitude on heating profiles

In this subsection we will focus on the smoke and non-smoke cases and examine to which extent variations in moisture and
aerosol loading affect the SW heating within the aerosol layer. Similarly, we will investigate if moisture, AOD and cloud top
altitude variations have a significant effect on the LW cooling rates at cloud top.

To account for differences in aerosol loading, each group of aerosol cases was divided into three AOD intervals (low, middle
and high). The intervals were chosen after taking all aerosol cases (smoke and non-smoke) from both periods (JJA and SO),
ranking their corresponding AOD values from lowest to highest, and then dividing them into three groups with the same number
of cases. Note that "same number of cases" is only valid when considering the full set of AOD values, i.e. for both periods and
all aerosol cases. Figure 6 shows that the average extinction in the aerosol layer always increases with increasing AOD while
there is no straight-forward relation between the AOD and average moisture ($RH$ or $q_v$) within the plume. During JJA, the
average RH and $q_v$ in the free troposphere increase with increasing AOD only for the smoke cases. For the non-smoke cases,
the middle AOD interval is the one with the highest moisture, particularly around 3 km. During SO, the highest AOD interval
is associated with the lowest $RH$ and $q_v$ ranges between approximately 2 and 6 km (for both smoke and non-smoke cases).
The Spearman's rank correlation coefficient (S) between AOD and the average $RH$ between cloud top and 7 km in the free
troposphere is positive (S=0.39 for the smoke and S=0.41 the for non-smoke cases) during JJA and negative in SO (S=-0.27
for the smoke and S=-0.22 for the non-smoke cases). The correlation values are statistically significant (p-value<0.05) for
both periods and both groups of cases. Similar statistically significant correlations are found between the AOD and $q_v$: S=0.44
(smoke) and S=0.43 (non-smoke) during JJA, and S=-0.21 (smoke) and S=-0.15 (non-smoke) in SO. Thus, for the data we
have used, a higher AOD does not necessarily imply higher moisture in the free troposphere at aerosol layer altitudes. This
result differs from findings made by Adebiyi et al. (2015) and Deaconu et al. (2019); we explore possible explanations for these
differences in Section 3.6. The radiative heating profiles for the highest and lowest AOD intervals together with the distributions
of maximum SW heating within the aerosol layer and LW cooling at cloud top are shown in Figure 7. The AOD values and
the aerosol type both have a distinct impact on the SW heating; the SW heating increases significantly with increasing AOD
and is higher for the smoke than for the non-smoke cases. In contrast, the LW cooling at cloud top does not show a clear and
general relation with the AOD level or the aerosol type. None of the cases or time periods show a clear difference in the average
potential temperature profiles between the three aerosol loading levels (Figure 6).

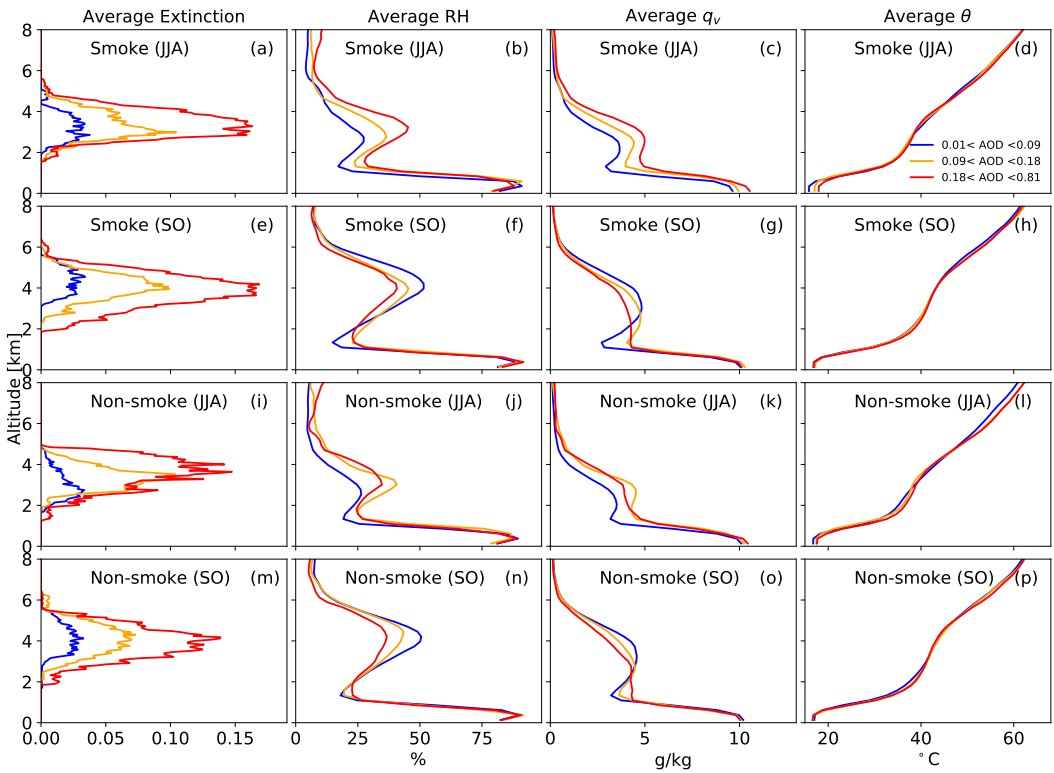

**Figure 6.** Mean profiles of aerosol extinction, relative humidity ($RH$), specific humidity ($q_v$) and potential temperature ($\theta$) for the smoke and non-smoke cases during the months June-July-August (JJA) and September-October (SO). $RH$, $q_v$ and $\theta$ are derived from atmospheric variables in the CloudSat ECMWF-AUX product. Cases are subdivided into three intervals as a function of the aerosol optical depth (AOD) value obtained from the CALIPSO Aerosol Profile Data Product.

To investigate the effects of the humidity of the aerosol layer on the atmospheric heating profiles we instead divide our aerosol cases into three intervals based on the average $RH$ and $q_v$ between cloud top altitudes and 7km. This altitude range was chosen to account for all the humidity within the altitudes corresponding to the aerosol layers as well as the humidity within in the vertical gap between aerosols and clouds. The average radiative heating profiles as well as the distributions of maximum SW heating within the aerosol layer and LW cooling at cloud top are shown for the intervals with high and low $RH$ (between cloud top altitudes and 7km) in Figure 8. The average profiles of extinction, $RH$ and temperature (for the three $RH$ intervals) are provided as supplementary information (Figure A2). A similar set of figures for the variable $q_v$ between cloud top and 7km is also provided in the supplement (Figures A3 and A4). There is no clear relationship between $RH$ and SW heating rates within the aerosol layer, since there is an increase in both variables during JJA for both cases (smoke and non-smoke) that does not occur during SO. An analysis of the relationship between AOD and moisture ($RH$ and $q_v$) during both periods and for both cases shows that the aerosol loading is the primary driver of the SW heating rates. The average LW cooling at cloud

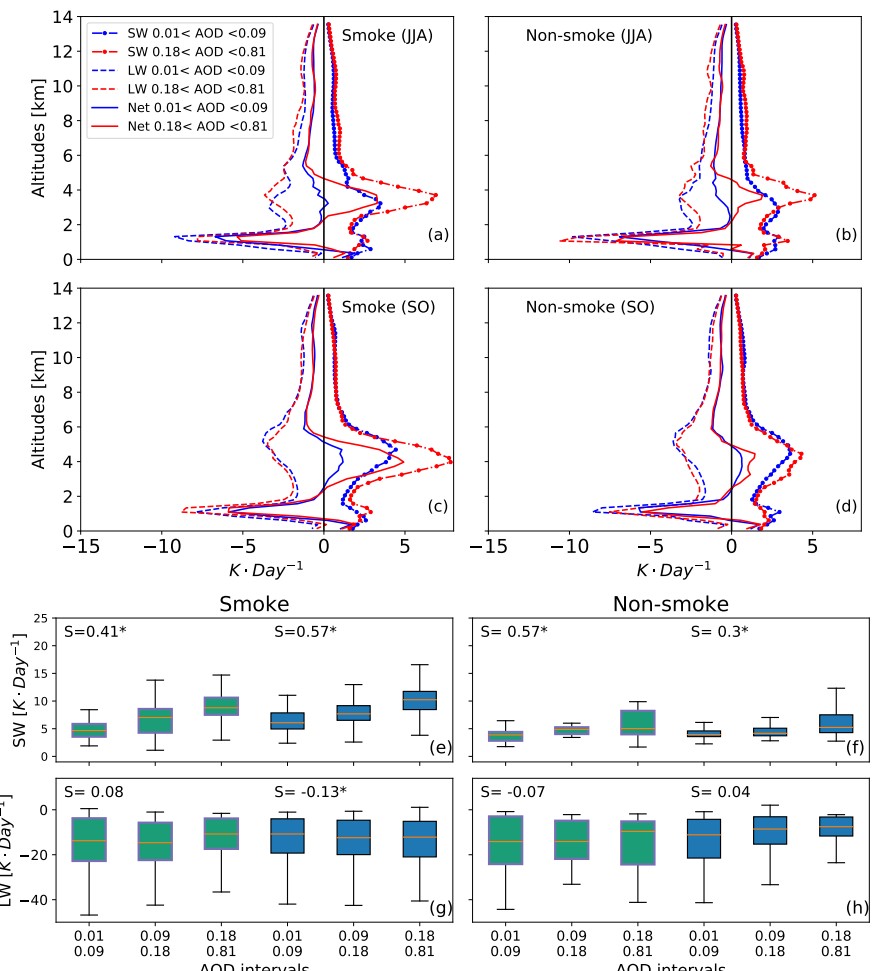

**Figure 7.** (a-d) Mean shortwave (SW), longwave (LW) and net (Net) radiative heating profiles for the smoke and non-smoke cases with high and low AOD during June-July-August (JJA) and September-October (SO) for the period 2007-2010. (e-h) Box plots of maximum SW heating within the aerosol layer (e-f) and maximum LW cooling at cloud top (g-h) for three AOD intervals for the smoke and non-smoke cases. Green boxes correspond to JJA and blue boxes to SO. For each case and period the Spearman's rank correlation coefficient (S) between the full range of AOD values and the SW (or LW) fluxes appears on top. Significant correlations (with p-value $< 0.05$) are marked with a star (*).

top decreases slightly (less cooling) with increasing $RH$ within the aerosol layer during JJA for the smoke cases, but this is not observed during SO. The Spearman's rank correlation values are also low for both periods suggesting a small influence. A

similar behavior is observed if $q_v$ is used instead of $RH$. (Figures A3 and A4)

The LW cooling at cloud top is inevitably dependent on cloud top altitude (CTA) as the cloud top temperature is strongly linked to the CTA. Figure 9 shows that CTA variations explain an important part of the variability of the cloud top LW cooling

for all the intervals of AOD, $RH$ and $q_v$, making it difficult to isolate a signal of the moisture and AOD impact on the cloud top radiative cooling. Note that in Figure 9, both periods and both aerosol cases have been combined in order to obtain a sufficient number of data points for each interval. Finally, we also segregated our cases into intervals of CTA in an attempt to see a relation between this variable and the AOD and/or the moisture. Figure A5 shows that there is not a general relationship between the CTA and the average AOD or humidity ($RH$ and $q_v$ between cloud top and 7 km) values for the different cases and periods. Furthermore, the S-values between the series of CTA and AOD, CTA and $RH$, and CTA and $q_v$ (not shown) are in general low. On the other hand the relationship between CTA and LW cooling is clear (as expected): in figure A6 the average radiative heating profiles (Figure A6 a-d) and the boxplots of CTA vs maximum LW cooling at cloud top (Figure A6 g-h) show an increase of the cloud top LW cooling with the CTA.

### 3.6 Comparison to previous studies and sensitivity to selection criteria

In this section, we summarize some similarities and differences in methods and results between our work and the related studies by Adebiyi et al. (2015) and Deaconu et al. (2019). Our decision to split the analysis into two periods was based on the results obtained by Deaconu et al. (2019). They found significant differences in moisture, wind speed, and wind direction in the free troposphere between JJA and SO over a similar study area. Our results support the findings by Deaconu et al. (2019) and Adebiyi et al. (2015) that the average moisture in the free troposphere (at aerosol layer altitudes) is higher during SO than during JJA and that the wind blowing from the continent is stronger during SO. Furthermore, we find that the altitude of the aerosol layers is higher in SO compared to JJA, which is in qualitative agreement with the results obtained by Deaconu et al. (2019) who compared the periods May-July and August-October. The difference in altitude can be explained by the strengthening and migration in altitude of the Southern African Easterly Jet during SO (Adebiyi et al., 2015; Adebiyi and Zuidema, 2016). In section 3.3 we noticed that our pristine cases have, on average, less moisture in the free troposphere than our aerosol cases (Figure 5a-c). Adebiyi et al. (2015) and Deaconu et al. (2019) obtained a similar result when comparing pristine (or low aerosol loading) and polluted (or high aerosol loading) situations. However, when dividing our aerosol cases into AOD intervals (section 3.5) we did not find a consistent increase in the average free tropospheric moisture (at aerosol layer altitudes) with increasing AOD, which contrasts the findings by Adebiyi et al. (2015) and Deaconu et al. (2019). Below we discuss possible explanations for this difference. We start by reviewing some details of Adebiyi et al. (2015) and Deaconu et al. (2019).

Adebiyi et al. (2015) combined sounding data from St. Helena Island (which is outside our area of our study) with the MODIS clear-sky fine-mode aerosol optical depth ($\tau_{af}$) for the period September-October from 2000 to 2011. They analyzed the relationship between the daily averaged $\tau_{af}$ and the moisture in the free troposphere by dividing the data into terciles of $\tau_{af}$. The intervals $\tau_{af} \leq 0.1$, $0.1 < \tau_{af} \leq 0.2$ and $\tau_{af} > 0.2$ where referred to as "pristine", "intermediate" and "polluted", respectively. In the present study, we have used a different dataset from a different source (CALIPSO and Cloudsat satellites) to analyze a different area during four (2007-2010) of the years studied by Adebiyi et al. (2015). Since the study areas are different, then meteorology may be different. It is also possible that we have differences in the hygroscopicity of the aerosols in the different areas. In Adebiyi et al. (2015), the study region is further from the continent compared to our work, i.e. aerosols that

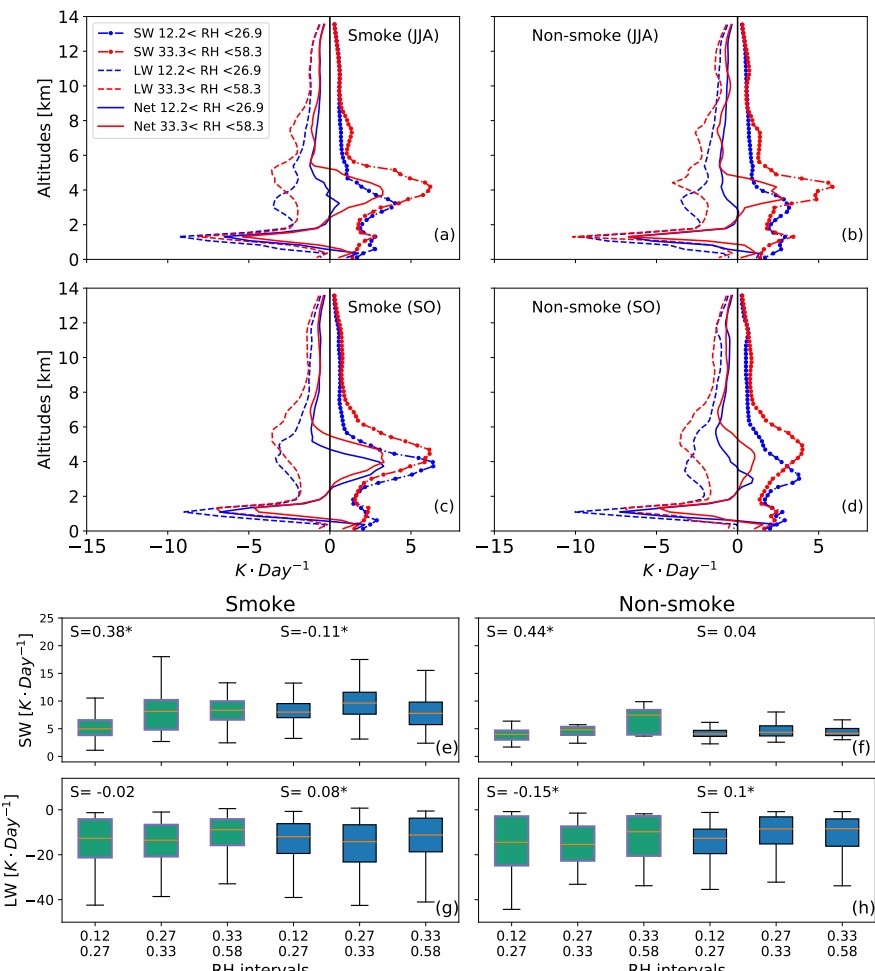

**Figure 8.** (a-d) Mean shortwave (SW), longwave (LW) and net (Net) radiative heating profiles for smoke and non-smoke cases with high and low $RH$ (in %) during June-July-August (JJA) and September-October (SO) of the period 2007-2010. (e-h) Box plots of SW heating in the aerosol layer (e-f) and LW cooling at cloud top (g-h) for three $RH$ intervals (values given in fraction). Right (left) panels correspond to the smoke (non-smoke) cases. Green boxes correspond JJA and blue boxes to SO. For each case and period the Spearman correlation coefficient (S) between the full range of $RH$ values and the SW (or LW) fluxes appears on top. Significant correlations (with p-value $< 0.05$) are marked with a star (*). $RH$ values are derived from variables in the CloudSat ECMWF-AUX product.

originally consisted of soot may have had time to become aged and more hygroscopic; this may affect the AOD-atmospheric moisture relation.

Deaconu et al. (2019) focused on the period from June to August of year 2008 to analyse situations with aerosols above clouds over an area very similar to the one we have analyzed in our work. They used CALIOP V3 products mainly to determine the vertical extension of the aerosol layers and they did not apply restrictions regarding the vertical separation between aerosols

and cloud layers. They used the values of aerosol optical thickness (AOT) at 865 nm retrieved by the POLDER (Polarization and Directionality of Earth Reflectances) instrument on board the PARASOL (Polarization and Anisotropy of Reflectances for Atmospheric Science coupled with Observations from a Lidar) satellite and compared two different aerosol-loading situations ($AOT > 0.04$ and $AOT < 0.01$ denominated as "high" and "low" respectively) in terms of cloud properties and meteorological parameters. They obtained the meteorological parameters from ERA-Interim (Dee et al., 2011; Berrisford et al., 2011). We have methodological differences compared to Deaconu et al. (2019); we imposed additional restrictions to select our aerosol above cloud profiles (using CALIOP V4), analyzed a longer period, separated situations with smoke aerosols from those with other kind of aerosols and separated our aerosol cases in intervals of AOD, moisture ($RH$ and $q_v$) in the free troposphere, and CTA.

We performed additional analysis to investigate if the restrictions used in the selection of our aerosol cases (both smoke and non-smoke) could play a role in the different relationship we found between AOD and free tropospheric moisture compared to Adebiyi et al. (2015) and Deaconu et al. (2019). Atmospheric profiles with aerosol extinction between 2 and 7 km were selected based on CALIPSO's Aerosol Profile Data Products. We did not require that clouds should be present below the aerosol layer(s) as we focus on the relationship between AOD and free tropospheric moisture. If clouds were present, we did not apply any restrictions (e.g. no limitation based on separation between aerosol and cloud layers, number of cloud layers or cloud top altitudes). Thus, profiles can exist where there are only aerosols, where aerosols and clouds are in contact, and where aerosols and clouds are separated. The predominant situation in this specific area should be that aerosols are above and separated from low level clouds (Deaconu et al., 2019). We also used the values of temperature, pressure and $RH$ derived from MERRA-2 that are contained in CALIPSO's Aerosol Profile Data product. Furthermore, we divided our profiles based on the AOD in three different ways (compare AOD intervals in Figure A7a-d with Figure A7e-h and Figure A7i-j). The AOD interval limits were chosen to resemble those used by Adebiyi et al. (2015), Deaconu et al. (2019), and the current work, respectively. Note, however, that the AOD values in the three original studies are from different sensors and that Deaconu et al. (2019) used the wavelength 865 nm. Thus, we do not an attempt to faithfully reproduce the previous studies.

For the period JJA, Figure A7 shows that the average moisture ($RH$ and $q_v$) in the free troposphere (between 2-7 km) and the temperature in the altitude range of the top of the inversion layer increase with increasing AOD and average aerosol extinction, independently of how we select the AOD intervals. Table 3 also shows that the Spearman correlation coefficients between the AOD and the average $RH$, $q_v$, and temperature (at inversion layer altitudes) are positive and significant. These results are in agreement with Adebiyi et al. (2015) and Deaconu et al. (2019). However, during SO, there is a clear change in the relationship between AOD and moisture, which warrants a separate analysis of each month. As for JJA, there is a positive and significant, albeit small, correlation between AOD and average moisture ($RH$ and $q_v$) during October. However, during September the variables are uncorrelated (Table 3). The correlation between AOD and temperature remains positive and significant for both months (September and October). Furthermore, Figure A8 shows that during September, the selection of the AOD intervals can determine, to some extent, if we observe a steady increase of the average moisture (between 2-7 km) with AOD (e.g. Figure A8 e-h) or not (e.g. Figure A8 a-d and i-l). On the other hand, the temperature profiles show an increase of the maximum temperatures (at the top of the inversion layer) with the increase of the average aerosol extinction regardless of the interval selection, which is in agreement with previous studies.

**Table 3.** Spearman correlation coefficient (S) between the AOD (Column Optical Depth Tropospheric Aerosols at 532nm from CALIPSO's Aerosol Profile Data product) and average moisture (RH and $q_v$) at the altitude range of aerosol layers in the free troposhpere (2-7 km) and between AOD and temperature (T) at the altitude range of the boundary layer top inversions (1.5-2 km ). Atmospheric profiles selected are those where there is aerosol extinction only between 2 and 7 km. Values are calculated during a four year period (2007-2010) for June-July-August (JJA), September and October. Correlation values marked with a star (*) are significant at the 95% level.

| Months | JJA | | | September | | | October | | |
|---|---|---|---|---|---|---|---|---|---|
| Moisture MERRA-2 | RH | $q_v$ | T | RH | $q_v$ | T | RH | $q_v$ | T |
| AOD (CALIPSO) | S= 0.48* | S= 0.50* | S=0.30* | S=0.0008 | S= 0.009 | S=0.27* | S= 0.19* | S= 0.18* | S=0.19* |

To summarize, the different results obtained in our study compared to Adebiyi et al. (2015) and Deaconu et al. (2019) may be due to a combination of different factors, such as the chosen study areas, periods analyzed, and different datasets or versions. Furthermore we do not discard the possibility of having an additional source of error due to AOD inaccuracies in CALIOP V4 or because $RH$ values are not measured directly by CALIPSO but extracted along the track from MERRA-2. In any case, a direct one-to-one comparison with the previous studies is difficult and some differences can therefore be expected.

## 4 Summary and conclusions

We have used CALIPSO and CloudSat retrievals for the years 2007-2010 to study situations when moist aerosol layers overlie low-level clouds over the Southeast Atlantic during the biomass burning season (June - October). We divided our data into two periods, June-July-August (JJA) and September-October (SO) to reduce the effect of seasonal meteorology changes on the studied aerosol-cloud interactions. Furthermore, we used the CALIOP V4 aerosol classification algorithm to separate cases with pristine air above clouds, smoke aerosols above clouds and other types of (non-smoke) aerosols above clouds.

The pristine cases displayed a clear difference in the large-scale wind pattern compared to the other two types of cases with aerosols above clouds. Easterly winds predominated in the smoke and non-smoke aerosol cases, which is also a prerequisite for bringing polluted continental air over the studied region, while the pristine cases were dominated by winds from the open ocean (cf. Fuchs et al., 2017; Deaconu et al., 2019). Consequently, it was not possible to conclude if any observed difference between the pristine and aerosol cases in low-level cloud properties (e.g. cloud top height) or thermodynamic properties of the atmosphere (e.g. stratification) were caused by the presence of an aerosol layer or by the differences in large-scale circulation. The two aerosol cases (non-smoke and smoke) displayed similar large-scale winds. They were both also associated with enhanced levels of moisture in the free troposphere, which is typical for biomass burning plumes that are advected from the continent (Haywood et al., 2003; Adebiyi et al., 2015; Deaconu et al., 2019). During JJA, a positive correlation between AOD and moisture was found in the free troposphere, in agreement with previous studies (e.g. Adebiyi et al., 2015; Deaconu et al.,

2019). In SO, we did not find a monotonous increase of the free tropospheric moisture with increasing AOD when using our specific selection criteria.

According to the CALIOP V4 aerosol classification algorithm, and in agreement with our expectations, smoke was the dominant aerosol type overlying the stratocumulus clouds during the biomass burning season. Nevertheless, a substantial amount of other kinds of aerosols were also detected within the pollution plumes. One explanation for the obtained result could be that the CALIOP algorithm misclassifies some of the smoke aerosols as other aerosols. Another explanation could be that other aerosol types than smoke indeed occasionally dominate the pollution plumes. Chazette et al. (2019) observed a mixture
of different aerosol types, mostly polluted dust and smoke, in the free troposphere over the coastal regions of Namibia (near the area of our study) during the biomass burning season. Their results are consistent with our findings and merits a broader definition of the pollution plumes overlying the stratocumulus clouds.

Our analysis clearly showed that the SW heating of the aerosol layer increased with higher aerosol loading and that the heating rates were higher in the smoke cases compared to the non-smoke aerosol cases. Moisture changes ($RH$ and $q_v$) between
cloud top and 7 km altitude had no clear impacts on the SW heating rates. These results are in agreement with Yamaguchi et al. (2015) and Deaconu et al. (2019) who also found a negligible impact of the aerosol layer moisture on the SW heating rates. A semi-direct aerosol effect would be expected to generate a change in the thermodynamic structure of the atmosphere and the cloud top height, but such an effect is not evident in the analyzed data. Previous studies have suggested that there is a weak overall semi-direct effect of elevated smoke layers over the Southeast Atlantic and that the gap between the absorbing
aerosol layer and the underlying cloud must be small (less than 0.5 km) to detect a significant influence (Herbert et al., 2020; Costantino and Bréon, 2013; Adebiyi and Zuidema, 2018). Our results are thereby not in contradiction with these studies as we selected cases with a minimum distance of 0.75 km to avoid any potential contact between the aerosol layer and the cloud.

No impact of the aerosol loading or type on the cloud top radiative cooling rates was found. For smoke, this result is expected since smoke aerosols do not absorb in the LW part of the spectrum (Yamaguchi et al., 2015). We found no clear relationship
between the LW cooling rates at cloud top and the moisture ($RH$ or $q_v$) in the free troposphere, most likely due to the strong variability in the cooling rates. These were instead found to be more associated with variations in CTA. Deaconu et al. (2019) calculated that an increase in the water vapor content of the aerosol layer from "low" to "high" could dampen the net cloud top cooling by about 5 $Kday^{-1}$. This is a small number compared to the variability of the LW cooling rates found in our analysis within one single $RH$ or $q_v$ interval. It shows the difficulty of detecting the impact of moisture changes within the aerosol layer
on the underlying clouds and the need to carefully constrain the meteorology.

*Data availability.* CALIPSO products (CAL_LID_L2_05kmMLay-Standard-V4-20 and CAL_LID_L2_05kmAPro-Standard-V4-20) were obtained from the Atmospheric Science Data Center (ASDC) website: https://asdc.larc.nasa.gov/project/CALIPSO, last access: 19 October 2020. Cloudsat products 2B-FLXHR-LIDAR.P2_R04 and ECMWF-AUX PR_04 were obtatined from the Cloudsat Data Processing Center website: http://www.cloudsat.cira.colostate.edu/order-data, last access: 19 October 2020. ERA5 datasets were obtained from the Climate
Data Store website: https://cds.climate.copernicus.eu/cdsapp#!/dataset/reanalysis-era5-pressure-levels?tab=form, last access: 3 March 2021.

*Author contributions.* The original idea of the study came from AMLE with input from ABP, AD and FAM. ABP performed the data analysis and provided all visualizations and wrote the initial paper draft. AD provided assistance with the satellite retrievals. ABP, AMLE, FB and AD analysed the results. ABP wrote the paper with input and revisions from AMLE, FB and AD.

*Competing interests.* The authors declare that they have no conflict of interest.

*Acknowledgements.* This study was funded by the Swedish National Space Agency grant 16317. The data analysis was performed on resources provided by the Swedish National Infrastructure for Computing (SNIC) at the National Supercomputing Centre at Linköping University.

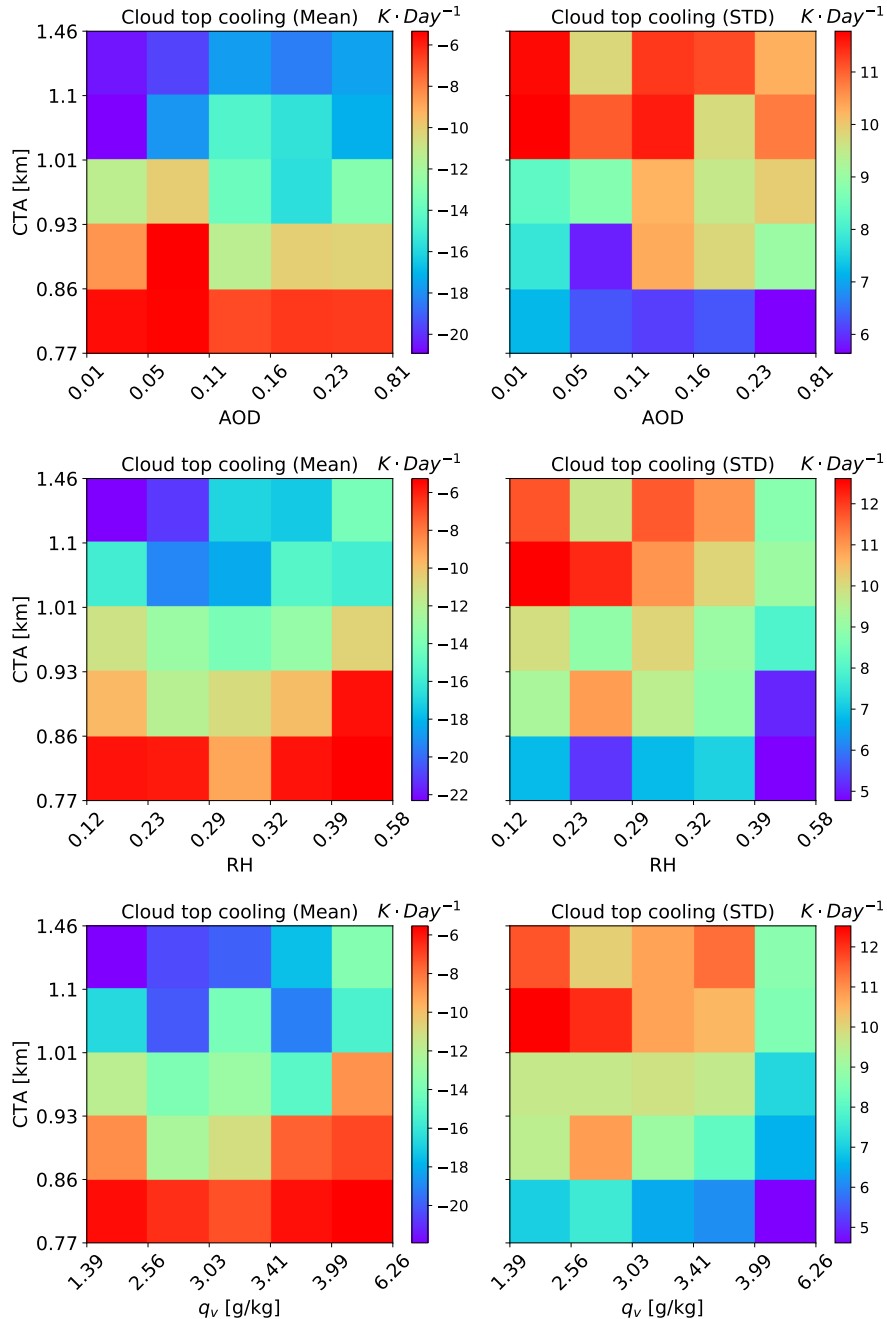

**Figure 9.** Histograms of the mean and the standard deviation (STD) of the minimum LW cooling at the cloud layer as functions of cloud top altitude (CTA) and AOD, CTA and $RH$ (between cloud top and 7km) and CTA and $q_v$ (between cloud top and 7km). Both periods (JJA and SO) and both aerosol cases (smoke and non-smoke) were used. $RH$ and $q_v$ values are derived from variables in the CloudSat ECMWF-AUX product.

# Appendix A: Supplementary material

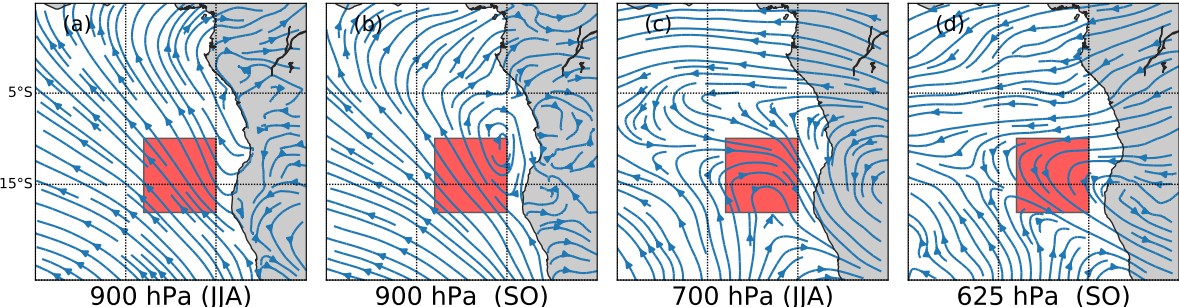

**Figure A1.** Streamlines corresponding to the average horizontal wind speed at 900 hPa (representative of cloud level) in JJA and SO, and 700 (625) hPa (representative of aerosol layer level) in JJA (SO) for the **Pristine\*** cases (i.e. profiles observed during days when no aerosol above cloud was detected along the satellite track within the whole area of study (red square).

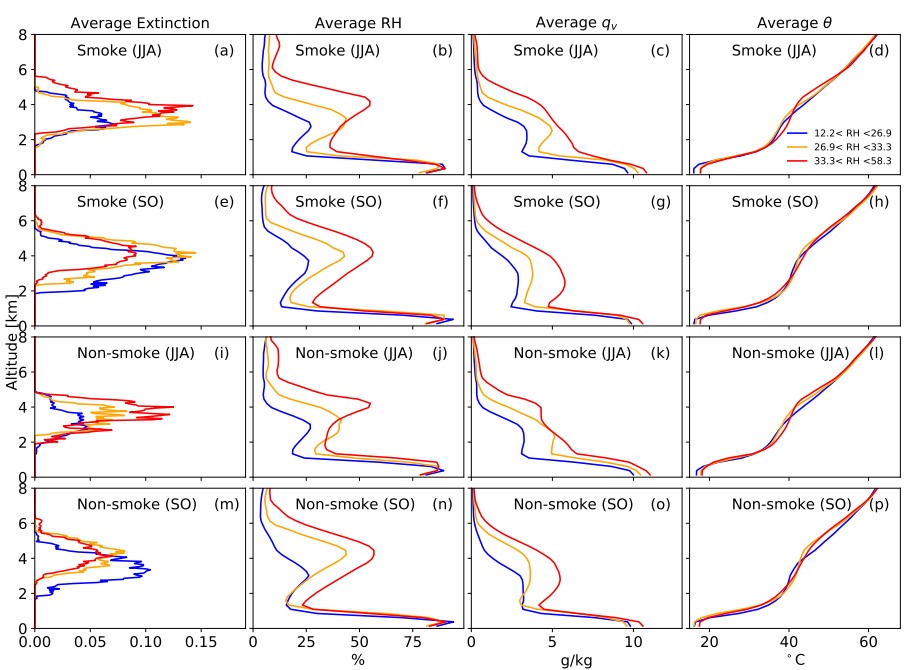

**Figure A2.** Mean profiles of aerosol extinction, relative humidity ($RH$), specific humidity ($q_v$) and potential temperature ($\theta$) for the smoke cases June-July-August (JJA) and September-October (SO). Cases where subdivided into 3 intervals depending on the RH value. $RH$, $q_v$ and $\theta$ values are derived from variables in the CloudSat ECMWF-AUX product.

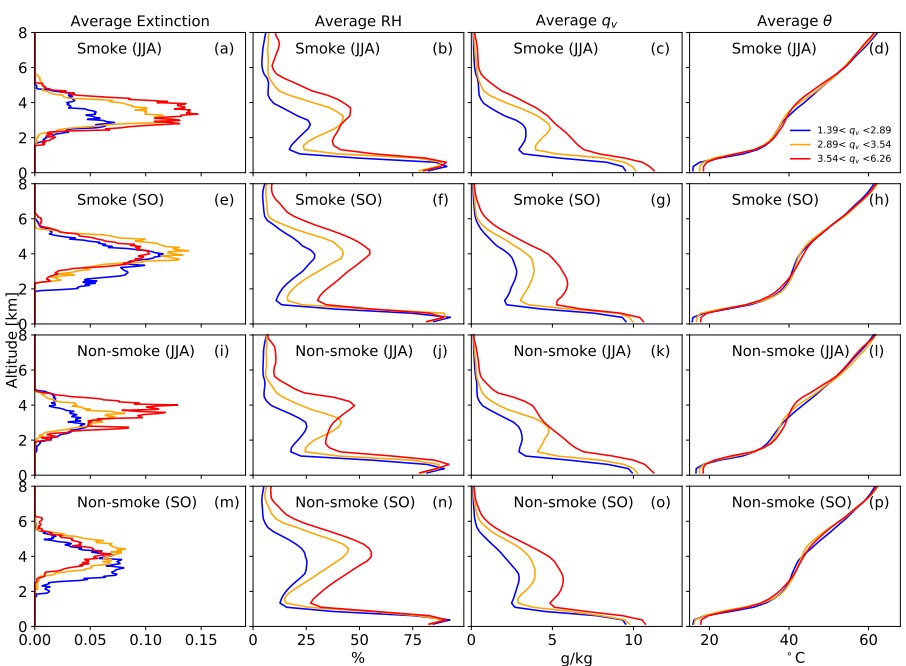

**Figure A3.** Mean profiles of aerosol extinction, relative humidity ($RH$), specific humidity ($q_v$) and potential temperature ($\theta$) for the smoke cases June-July-August (JJA) and September-October (SO). Cases were subdivided into three intervals as a function of the $q_v$ value. $RH$, $q_v$ and $\theta$ values are derived from variables in the CloudSat ECMWF-AUX product.

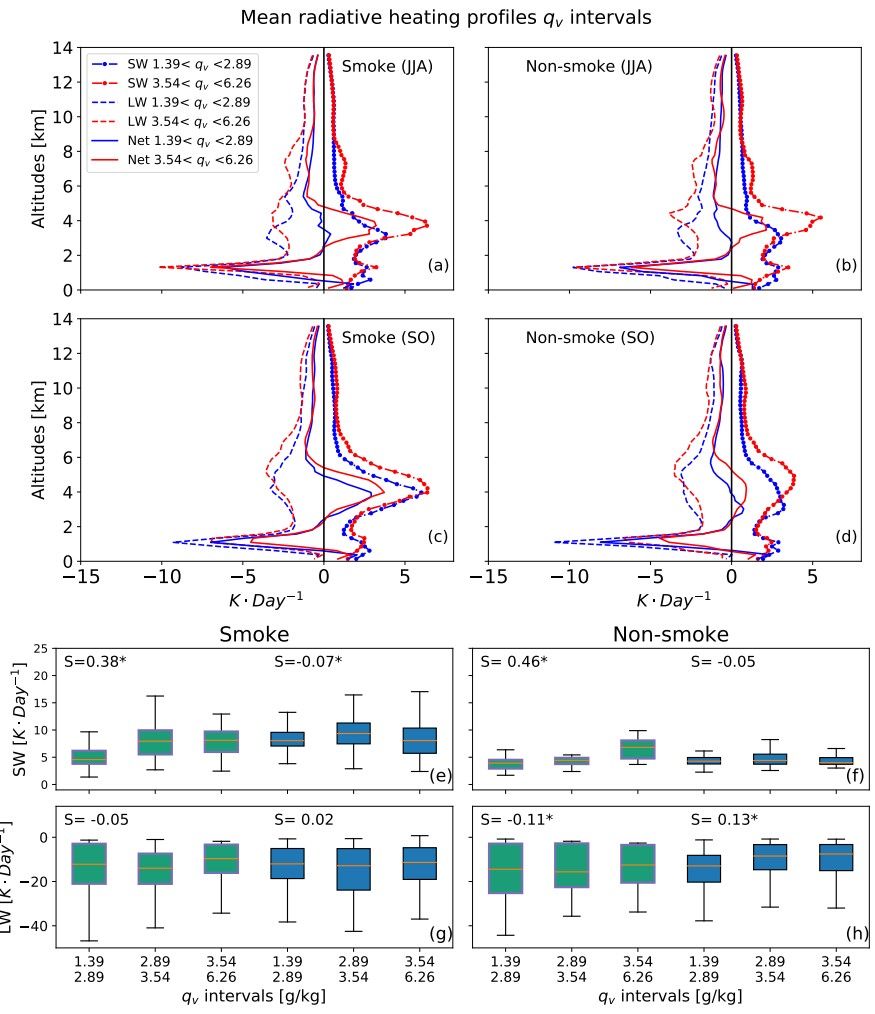

**Figure A4.** (a-d) Mean shortwave (SW), longwave (LW) and net (Net) radiative heating profiles for smoke and non-smoke cases with high and low RH June-July-August (JJA) and September-October (SO) of the period 2007-2010. (e-h) Box plots of SW heating in the aerosol layer (e-f) and LW cooling at cloud top (g-h) for three RH intervals. Right (left) panels correspond to the smoke (non-smoke) cases. Green boxes correspond JJA and blue boxes to SO. For each case and period the Spearman correlation coefficient (S) between the full range of $q_v$ values and the SW (or LW) fluxes appears on top. Significant correlations (with p-value $< 0.05$) are marked with a star (*). $q_v$ values are derived from the CloudSat ECMWF-AUX product.

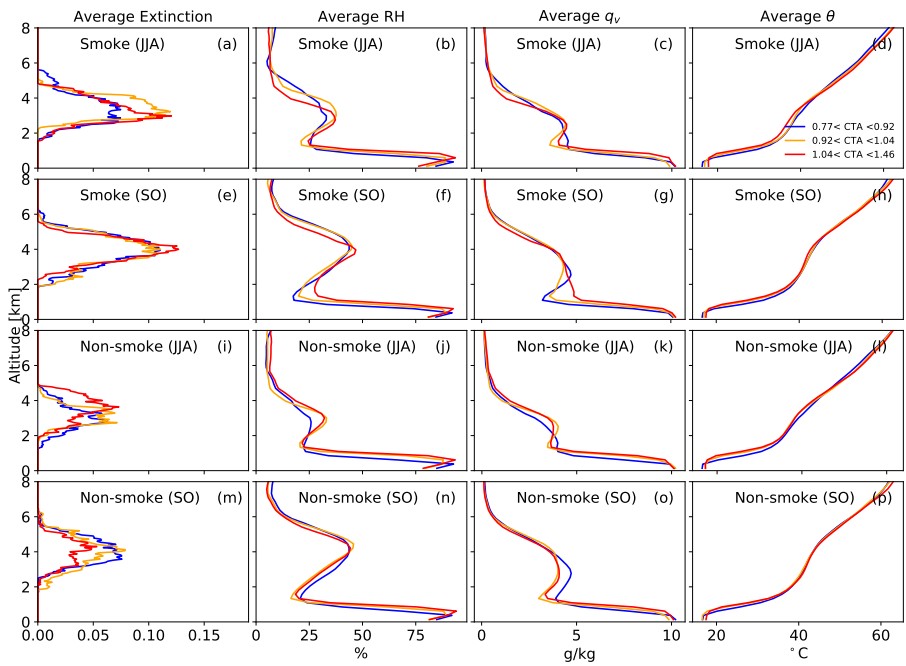

**Figure A5.** Mean profiles of aerosol extinction, relative humidity ($RH$), specific humidity ($q_v$) and potential temperature ($\theta$) for the smoke cases June-July-August (JJA) and September-October (SO). Cases were subdivided into three intervals as a function of the CTA value.

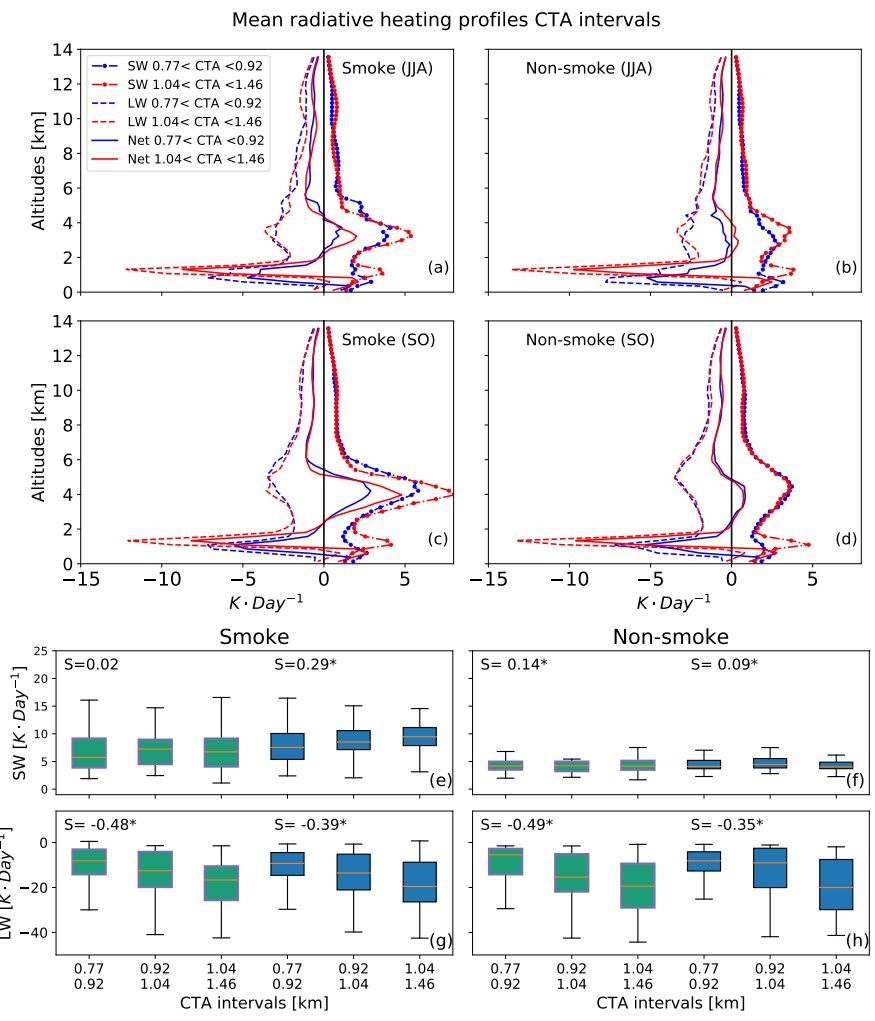

**Figure A6.** (a-d) Mean shortwave (SW), longwave (LW) and net (Net) radiative heating profiles for smoke and non-smoke cases with high and low RH June-July-August (JJA) and September-October (SO) of the period 2007-2010. (e-h) Box plots of SW heating in the aerosol layer (e-f) and LW cooling at cloud top (g-h) for three CTA intervals. Right (left) panels correspond to the smoke (non-smoke) cases. Green boxes correspond JJA and blue boxes to SO. For each case and period the Spearman correlation coefficient (S) between the full range of RH values and the SW (or LW) fluxes appears on top. Significant correlations (with p-value < 0.05) are marked with a star (*).

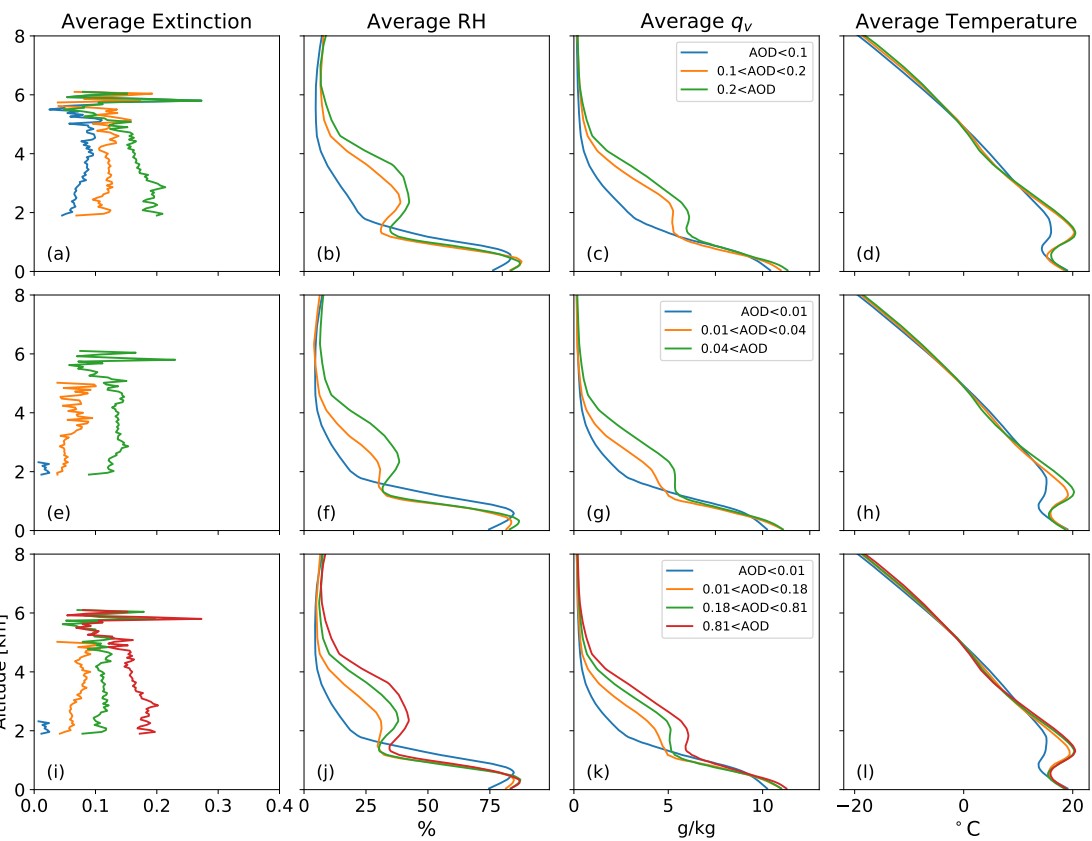

**Figure A7.** Mean profiles of aerosol extinction, relative humidity ($RH$), specific humidity ($q_v$) and temperature for CALIPSO profiles containing aerosols above clouds during de months June- July-August from 2007 to 2010. The data used for subfigures: a-d, e-h and i-l are exactly the same. The only difference is the selection of the intervals limits and interval number for which the mean values are computed (cf. 3.6). The aerosol optical depth (AOD) values are obtained from the CALIPSO Aerosol Profile Data Product. $RH$, $q_v$ and temperature are originally from MERRA-2 and added along the satellite track to the CALIPSO Aerosol Profile data product.

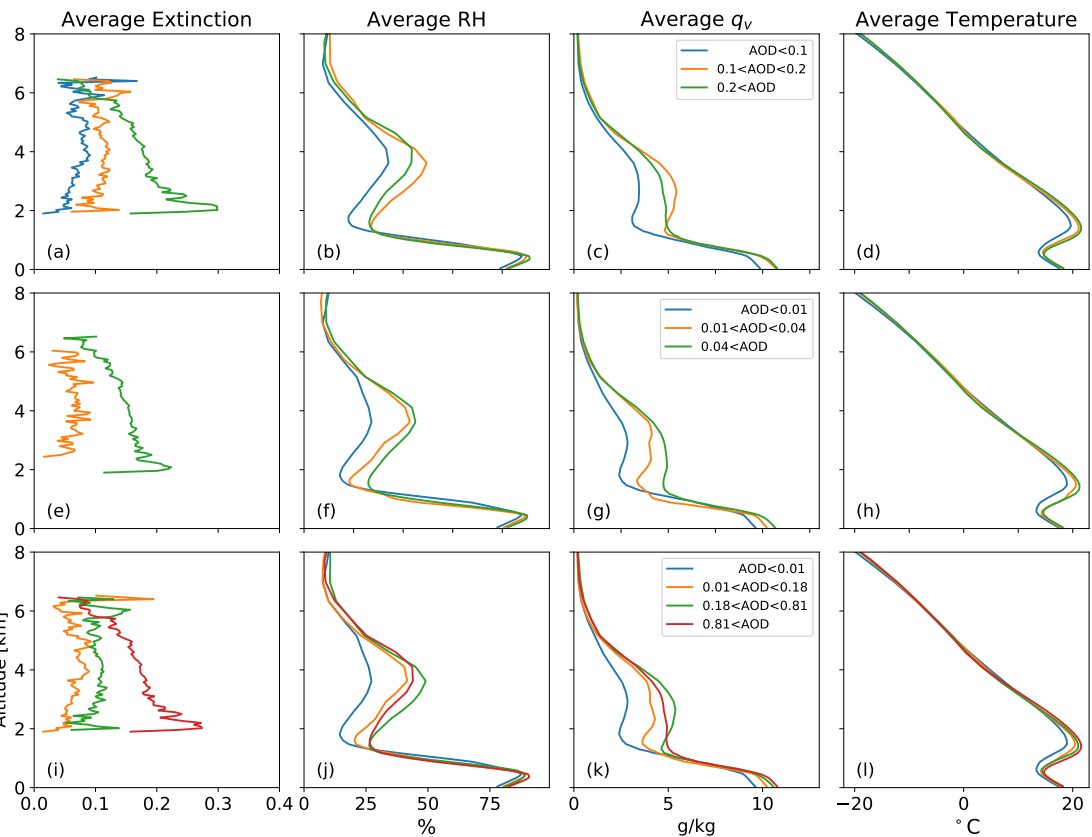

**Figure A8.** Mean profiles of aerosol extinction, relative humidity ($RH$), specific humidity ($q_v$) and temperature for CALIPSO profiles containing aerosols above clouds during de month of September from 2007 to 2010. The data used for subfigures: a-d, e-h and i-l are exactly the same. The only difference is the selection of the intervals limits and interval number for which the mean values are computed. The aerosol optical depth (AOD) values are obtained from CALIPSO Aerosol Profile Data Product. $RH$, $q_v$ and temperature are originally from MERRA-2 and added along the satellite track to the CALIPSO Aerosol Profile data product.

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
