# Peer review of "Impact of smoke and non-smoke aerosols on radiation and low-level clouds over the Southeast Atlantic from co-located satellite observations"

_Atmospheric Chemistry and Physics, 2020_

## Referee Comment (RC1) · Anonymous Referee #1 · 17 Nov 2020

The paper written by Baró Pérez et al. is exploring situations with moist aerosol layers above stratocumulus clouds in the Southeast Atlantic, during the biomass burning season. The authors attempt to separate and quantify the impacts of aerosol loading and type and humidity on the radiative fluxes. They employed observations from CALIOP and CloudSat satellites and meteorological parameters from MERRA-2 and ERA5 reanalysis, in order to analyse the meteorological effect (in the pristine cases) and the aerosol effect (polluted cases of different types of aerosols – smoke and mixed type) on the atmospheric heating rates, i.e. cloud top cooling. The paper makes reference

to the paper by Deaconu et al, 2019 which found that separating meteorology from aerosol effects on below clouds it is not achievable from observations, as the meteorology is substantially different between polluted and less polluted (clean) cases in the Southeast Atlantic. The differences in approach are that the present paper considers four years of satellite datasets (2007-2010), the distance between the clouds and the above aerosol layer is minimum 0.4 km and aerosols are classified based on their type. The authors divide their data into two periods, June-July-August (JJA) and September-October (SO) due to seasonal meteorological differences. Most of their findings are in agreement with previous studies: e.g the shortwave (SW) heating of the aerosol layer increased with higher aerosol loading, the relative humidity (RH) of the aerosol layer had a negligible impact on the SW heating rate, no impact of smoke on the underlying cloud top radiative cooling, and enhanced levels of moisture are transported within the aerosol plumes. However, they haven't found indication of a semi-direct effect of aerosols, or a relationship between aerosol loading and increased RH.

The paper is well documented and generally well written. The methodology is different from Deaconu et al., 2019, but following a similar train of thought, and their findings are mostly complementary. This paper adds value and interesting discussions to the subject of aerosols above clouds and their radiative impact on clouds. I am recommending this paper for publication after the following remarks are addressed.

Deaconu et al., 2017 showed that CALIOP V3 underestimates the AOD above clouds with a factor of 2 to 4 when compared to other methods dedicated for above-cloud aerosol retrievals. While CALIOP V4 has improved the calibration at 532 nm compared to V3, the AOD retrieval is still underestimated over ocean compared with MODIS, showing little improvement over the Southeast Atlantic Ocean (i.e. Fig 16, Kim et al 2018). In other words, the extinction coefficient and/or the aerosol layer geometrical thickness are underestimated for thick aerosol layers. How do the authors justify using this extinction coefficient to compute the radiative heating rates, without any additional scaling? Following, how do the authors justify their choice of 0.4 km between the cloud

top and the base of the aerosol layer (as detached cases), considering also that CATS system shows that the CALIOP V3 algorithm probably overestimates the base of the aerosol layer by 500 m (Rajapakshe et al., 2017) – which probably stands true for V4 as well.

The authors should mention that the sole difference between polluted continental/smoke and elevated smoke is their altitude separation and that pollution lofted by convective processes or other vertical transport mechanisms can be misclassified as elevated smoke (especially that the V4 lidar ratios used in the CALIOP retrieval algorithm are identical for both situations ($70\pm16$ sr at 532 nm and $30\pm14$ sr at 1064 nm)).

I am not entirely convinced by the choice of the three AOD intervals. I believe that using different thresholds for the AOD intervals for different regions and time periods is confusing. For example what you consider 'high' AOD interval for the mixed cases JJA is covering mostly the 'middle' AOD for the smoke cases SO. While a comparable number of profiles for each bin is fairly important for a statistical analysis, I find it more important to compare same ranges of values. I would suggest either reducing the number of intervals (above a threshold and below a threshold) or choosing values that are applicable to all categories (e.g. in the case of mixed type, have only 2 intervals – as it is clear the AOD values are lower than for smoke).

Is it possible that the different RH profiles between JJA and SO are associated to different meteorological conditions (e.g. wind direction from the ocean, instead from land)? Deaconu et al, 2019 looked at high and low AOD above clouds cases and found that easterlies (winds predominantly from the NE) are associated with larger AODs and larger humidity values, while the wind coming from the open ocean is characterized by low values of AOD and humidity. The author also mentions that the average monthly horizontal winds show a significant difference between the months of SO compared to JJA at 700hPa. In this paper the authors also separate AOD in different intervals and analyze the RH, but it is not clear if these cases have similar underlying meteorology,

and if for all these cases the aerosol and humidity layers are in fact together transported from the land. Considering that the CALIOP Aerosol Profile Data product includes the RH extracted along the track from MERRA-2, but no information on the winds, I suggest mentioning the caveats associated to this approach when comparing these cases and the different resulting heating rates.

The results showing mean profiles of relative humidity for the three aerosol intervals for SO period are surprising. Adebiyi et al., 2015 use radiosondes measurements from St. Helen for different aerosol loading intervals (e.g. Fig 11, Fig 14) and their results show both the humidity and temperature boundary layer-top inversions strengthen as the aerosol loading above increases. Flight campaign measurements that took place in the Southeast Atlantic during the biomass burning season (ORACLES, CLARIFY and AEROCLO-sA) have also found increased moisture with increased aerosol loading. Would the authors like to comment on this?

Fig.6: Could the authors also plot specific humidity profiles? Even in a supplementary figure.

The authors mention 'no indication of a semi-direct effect'. However, there is no additional study on the cloud liquid water path and cloud optical thickness that would support this comment. Their statement is based on the results on the LW cooling rates due to RH, which, as they mentioned, shows confounding impacts due to the variability of the cloud top cooling rates. This study would be enhanced with a sensitivity test, in which the cloud top altitude would be fixed for the different aerosol types, AOD and RH intervals and time periods.

---

## Referee Comment (RC2) · Anonymous Referee #2 · 28 Nov 2020

The study examines the impacts of smoke and other aerosols species on the low-level clouds over the southeast Atlantic. The paper's main goal is to characterize the effects of aerosol loading, aerosol types, and the mid-tropospheric relative humidity on the thermodynamical profile, radiative heating rates, and the cloud-top radiative cooling. To do so, the study used aerosol and cloud retrievals from CALIPSO and CloudSat as well as MERRA-2-based meteorological variables obtained as part of CALIPSO and ERA5 reanalysis dataset (see Table 1). The study focused on the area between 10 to 18°S and 2 to 10°E, and between June and October 2007 and 2010. According to

the study, the justification of the area and study period is because the low-level cloud and above-cloud aerosol maximize sometimes between June and October. In addition, the authors also added that the months of study were chosen in other to compare their results with previous studies of (Deaconu et al., 2019) and (Adebiyi et al., 2015), who used June-August and July-October period, respectively. Furthermore, the study divided their dataset into three cases: (1) the smoke cases – based on the aerosol retrieval classified as "elevated smoke" in CALIPSO. These are cases where the cloud layers (between 0.75 km and 2.5 km) are separated from the "elevated smoke" layers by at least 0.4 to 6 km. (2) The mixed cases are the same as the smoke cases, except that they are for other CALIPSO aerosol retrievals NOT classified as "elevated smoke". (3) The pristine cases with cloud layers between 0.75km and 2.5km but no aerosol above.

The study finds that the smoke and mixed cases have similar meteorological conditions (winds direction, thermodynamical profile), although with different magnitudes. The study also suggests that no monotonous increase in the mid-tropospheric relative humidity with increasing aerosol optical depth exists. In addition, they find that while the vertical distribution of the SW heating rates corresponds to the above-cloud aerosol extinction for both the smoke and mixed cases, they do not correlate with the moisture distribution.

Given the importance of the southeast Atlantic's aerosol-cloud interaction to the global radiative budget, this article will add to the growing body of knowledge about the complicated aerosol-cloud-meteorology system over the southeast Atlantic. While the article is generally well written, I have some comments that I hope the authors will address.

1. The authors should change the title. While smoke is undoubtedly an absorbing aerosol, I don't think the authors have shown sufficiently argued that the "mixed cases" are indeed "non-absorbing" aerosols. To my understanding, these mixed cases also contain smoke and dust, which are both absorbing aerosols (Samset et al., 2018).

2. That said, I think calling other CALIPSO-derived aerosol types that do not fall within the "elevated smoke" classification "mixed cases" might be somewhat inaccurate. The reason is that the term "mixed" gives the impression there are always more than one aerosol species within the aerosols layer at all times, different from the elevated smoke. Although Fig. 1 shows cases of polluted/continental smoke, dust, and dusty marine aerosols, it is unclear whether those cases represent the entire atmospheric column or only the mid-troposphere. Moreover, the lidar ratio used to separate the elevated smoke in CALIPSO V4 is identical to that of polluted smoke, except that elevated smoke is defined as smoke in the mid-troposphere. Hence, if indeed the authors focused on aerosols above clouds, then part of the polluted smoke included with the mixed cases is in-fact misclassified elevated smoke in the mid-troposphere, which is expected to have similar optical properties, and therefore similar impacts on meteorology and low-level clouds as the elevated smoke. Hence, in addition to changing the classification name, the authors need to make a stronger case that the optical properties of the "mixed cases" are in fact, different from those of "smoke cases". The author should also show that a significant amount of dust and smoke are represented at all height levels of the mid-troposphere to justify their "mixed cases" argument. Perhaps Fig. 3 can be done for the different aerosol species identified in Fig. 1.

3. The author should discuss the differences between this study and that of (Adebiyi et al., 2015) and (Deaconu et al., 2019), given that the conclusions in this study are sufficiently different from the other two studies (to some degree). Although some comparisons were made in some parts of the texts, it is mostly incoherent and unclear to the reader looking to know how exactly this study stands out from the previous papers. I will suggest that the authors have a separate sub-section, where they discuss specifically the differences between this study and previous ones.

4. The authors did not explore the caveat that CALIPSO bottom layers have substantial uncertainty, as have been shown in previous studies. For example, CALIPSO has been shown to have a significant bias when compared to aircraft-based observation from

HSRL (e.g. Kacenelenbogen et al., 2011), and satellite-based observation from CATS (e.g. Rajapakshe et al., 2017)). The authors should account for this uncertainty in their analysis. In addition, the authors should also discuss how this caveat could affect their conclusions.

5. Surprisingly, the authors find decreasing RH as a function of AOD. First, as indicated below, I think the same AOD and RH intervals should be used for all cases highlighted in Fig.6-8 ). That puts the results on equal footing and allow for better comparison. Secondly, I wonder if something is inherently mistaken in the dataset that the authors used, since it is wildly different from the previous study. Adebiyi et al. 2015 used radiosonde collected at St. Helena Island and found a reasonable relationship between AOD and moisture in the mid-troposphere. Other studies that used reanalysis datasets have found a similar positive relationship. Here, the authors used MERRA temperature and relative humidity values packaged with CALIPSO retrievals, not those directly obtained from the MERRA website. While I do not suggest that these datasets may be different, I think the author should validate their results using other reanalysis datasets. Specifically, the authors should use the ERA (and possibly NCEP) reanalysis and show that the same result holds as those found in this study. Those results/plots should be included in the supplementary document and cited in the paper. It is worth noting that these reanalysis datasets are often significantly different, especially over the ocean where limited ground-based constraints are assimilated. Nevertheless, as Adebiyi et al. 2015 (see their Fig. 14) showed, the reanalysis datasets are expected to be broadly consistent. Third, I wonder if other criteria set in the analysis also do play a role in the differences? The authors could use their datasets and test their results over the same regions and periods defined in other studies (Adebiyi et al., 2015; Deaconu et al., 2019). Finally, given that this part of the study is in sharp contrast from other studies, the authors must spend a sufficient amount of time explaining to the reader that these results are accurate, and also discuss in detail the difference between this study and others that found different results (see comment above).

6. Lastly, the details of all dataset should be discussed in the methodology. In particular, a lot of this study's results rely on the CloudSat-derived radiative fluxes. While those radiative fluxes may have used CALIPSO aerosol extinction profiles, they make other important assumptions about the aerosol's optical properties that can have important implications on this paper's conclusion. The authors must discuss the limitation of those assumptions in their results. In addition, there is a potential difference in the thermodynamical profiles used to obtain those CloudSat-derived fluxes (ERA) and those used for general characterization of meteorology in this study. As suggested above, the author must show that both reanalysis datasets support their overall conclusion.

Other Comments

1. Section 2.1: Given that the study relies on elevated smoke versus other pollution, some more details should be provided on how CALIPSO made those classifications.

2. Line 169: I am finding it difficult to reconcile why the number of profiles shown in Table 2, for smoke cases, for example, is higher than the total number of cases of elevated smoke shown in Fig. 1. I understand that the "smoke cases" should be a subset of the CALIPSO's "elevated smoke" classification since it has some additional criteria. The same discrepancy appears to be present in mixed cases as well. Are they strictly cases where the aerosol layer is above the low-level cloud for both the smoke and mixed cases? That is what Section 2.3 appears to suggest. The difference between Fig. 1 and Table 2 should be further clarified.

3. Line 175-178: Based on previous studies, like some of those cited by the authors, I think it is unlikely to have more cases of aerosols above clouds between 2-6E than 6-10E. I will suggest that the authors make the same plot as Fig. 3 but separated as a function of longitude or latitude. Such a figure can be included in the supplementary document.

4. Figure: While it may already be mentioned in the text, Fig. 2 should include the latitude range considered for the longitude distribution. Similar thing for the latitude

distribution. Also, for Fig. 3.

5. Line 208: Change ". . .southerly component compared to the two aerosol cases" to ". . .southerly winds compared to the two aerosol cases."

6. Line 227: Please re-write the sentence.

7. Line 214 should be Figure 5a-c

8. Line 231 should be Figure 5d–?

9. Section 2.1 There should be a brief description of how the radiative heating rate of Cloudsat is obtained to give the reader the complete picture and the ingredients to interpret the result adequately. For example, while Cloudsat may have used the vertical extinction profile from CALIPSO, what spectral distribution of single scattering albedo and asymmetric factor are assumed? What surface parameters that could affect radiative heating is assumed? How do they treat other wavelengths that are not provided by CALIPSO in longwave and shortwave? The author needs to provide a complete context to interpret the result.

10. Fig. 5 (as on others, see comments above) – always provide complete information for each figure. In this case, information of what data is plotted should be mentioned in the comment section.

11. Fig 6: Again, information needs to be complete. Where is the AOD coming from?

12. Like 249-254: The two paragraphs are repeated. The author should carefully go through the manuscript before any future re-submission.

13. Line 256-58: Any comparison between different cases will likely not be "fair" – that is, it will result in comparing "oranges" against "apples". I will suggest that the authors make the tercile range the same for the AOD classes.

14. Section 3.5: If AOD will be used to make inference about RH with the aerosol plume, it must be AOD of the aerosols above the cloud. It is difficult to expect a clear

linear relationship between RH averaged within a layer and column integrated AOD.

15. Line 275: Is this RH averaged within the aerosol layer or the entire atmospheric column? Please clarify this within the text and in the figure caption.

16. Fig A1: Check the caption. AOD or RH?

References Adebiyi, A. A., Zuidema, P. and Abel, S. J.: The Convolution of Dynamics and Moisture with the Presence of Shortwave Absorbing Aerosols over the Southeast Atlantic, J. Clim., 28(5), 1997–2024, doi:10.1175/JCLI-D-14-00352.1, 2015. Deaconu, L. T., Ferlay, N., Waquet, F., Peers, F., Thieuleux, F. and Goloub, P.: Satellite inference of water vapour and above-cloud aerosol combined effect on radiative budget and cloud-top processes in the southeastern Atlantic Ocean, Atmos. Chem. Phys., 19(17), 11613–11634, doi:10.5194/acp-19-11613-2019, 2019. Kacenelenbogen, M., Vaughan, M. A., Redemann, J., Hoff, R. M., Rogers, R. R., Ferrare, R. A., Russell, P. B., Hostetler, C. A., Hair, J. W. and Holben, B. N.: An accuracy assessment of the CALIOP/CALIPSO version 2/version 3 daytime aerosol extinction product based on a detailed multi-sensor, multi-platform case study, Atmos. Chem. Phys., 11(8), 3981–4000, doi:10.5194/acp-11-3981-2011, 2011. Rajapakshe, C., Zhang, Z., Yorks, J. E., Yu, H., Tan, Q., Meyer, K., Platnick, S. and Winker, D. M.: Seasonally Transported Aerosol Layers over Southeast Atlantic are Closer to Underlying Clouds than Previously Reported, Geophys. Res. Lett., (410), doi:10.1002/2017GL073559, 2017. Samset, B. H., Stjern, C. W., Andrews, E., Kahn, R. A., Myhre, G., Schulz, M. and Schuster, G. L.: Aerosol Absorption: Progress Towards Global and Regional Constraints, Curr. Clim. Chang. Reports, 4(2), 65–83, doi:10.1007/s40641-018-0091-4, 2018.

---

## Author Comment (AC1) · 18 Feb 2021

Reply to referee #1 comments on manuscript:

The paper written by Baró Pérez et al. is exploring situations with moist aerosol layers above stratocumulus clouds in the Southeast Atlantic, during the biomass burning season. The authors attempt to separate and quantify the impacts of aerosol loading and type and humidity on the radiative fluxes. They employed observations from CALIOP and CloudSat satellites and meteorological parameters from MERRA-2 and ERA5 re-

analysis, in order to analyse the meteorological effect (in the pristine cases) and the aerosol effect (polluted cases of different types of aerosols – smoke and mixed type) on the atmospheric heating rates, i.e. cloud top cooling. The paper makes reference to the paper by Deaconu et al, 2019 which found that separating meteorology from aerosol effects on below clouds it is not achievable from observations, as the meteorology is substantially different between polluted and less polluted (clean) cases in the Southeast Atlantic. The differences in approach are that the present paper considers four years of satellite datasets (2007-2010), the distance between the clouds and the above aerosol layer is minimum 0.4 km and aerosols are classified based on their type. The authors divide their data into two periods, June-July-August (JJA) and September-October (SO) due to seasonal meteorological differences. Most of their findings are in agreement with previous studies: e.g the shortwave (SW) heating of the aerosol layer increased with higher aerosol loading, the relative humidity (RH) of the aerosol layer had a negligible impact on the SW heating rate, no impact of smoke on the underlying cloud top radiative cooling, and enhanced levels of moisture are transported within the aerosol plumes. However, they haven't found indication of a semi-direct effect of aerosols, or a relationship between aerosol loading and increased RH.

The paper is well documented and generally well written. The methodology is different from Deaconu et al., 2019, but following a similar train of thought, and their findings are mostly complementary. This paper adds value and interesting discussions to the subject of aerosols above clouds and their radiative impact on clouds. I am recommending this paper for publication after the following remarks are addressed.

We thank the reviewer for the encouraging remarks and constructive comments.

1- Deaconu et al., 2017 showed that CALIOP V3 underestimates the AOD above clouds with a factor of 2 to 4 when compared to other methods dedicated for above-cloud aerosol retrievals. While CALIOP V4 has improved the calibration at 532 nm compared to V3, the AOD retrieval is still underestimated over ocean compared with MODIS, showing little improvement over the Southeast Atlantic Ocean (i.e. Fig 16, Kim

et al 2018). In other words, the extinction coefficient and/or the aerosol layer geometrical thickness are underestimated for thick aerosol layers. How do the authors justify using this extinction coefficient to compute the radiative heating rates, without any additional scaling? Following, how do the authors justify their choice of 0.4 km between the cloud top and the base of the aerosol layer (as detached cases), considering also that CATS system shows that the CALIOP V3 algorithm probably overestimates the base of the aerosol layer by 500 m (Rajapakshe et al., 2017) – which probably stands true for V4 as well.

To our knowledge, for CALIOP V4, there is not yet a similar study to that one done by Deaconu et al. (2017) where CALIOP V3 was used. Therefore there is no clear procedure for correcting the AOD/extinction values and we do not want to use an arbitrary correction factor. Based on Deaconu et al. (2017) and Jethva et al. (2014), the underestimation problem is more pronounced for high values of AOD above clouds in CALIOP V3. It is true that Fig16 in Kim et al 2018 (albeit with a very coarse resolution) shows little improvement over the Southeast Atlantic for CALIOP V4; however there is no detailed analysis about the uncertainties or biases in V4 for the aerosols above cloud situations in this area. Thus we can only recognize that this problem is a potential source of uncertainty also in V4. This has been added to Section 2.1.

We recognize that, according to Rajapakshe et al. (2017), 0.4 km can be a small distance for ensuring that aerosols and cloud layers are vertically separated. We thank the reviewer for pointing this out and have now increased the distance to 0.75 km, which is the same value used by Constantino and Breon (2013) in their '"well separated cases" (aerosol layers separated from cloud layers). With an increased distance, the number of "aerosol above cloud" cases decreased and all the figures of the manuscript have been remade with this reduced dataset.

2- The authors should mention that the sole difference between polluted continental/ smoke and elevated smoke is their altitude separation and that pollution lofted by convective processes or other vertical transport mechanisms can be misclassified as elevated smoke (especially that the V4 lidar ratios used in the CALIOP retrieval algorithm are identical for both situations (70±16 sr at 532 nm and 30±14 sr at 1064 nm)).

There was a mistake with respect to the aerosol classification in Fig.1 in the previous version of the manuscript. There are no "polluted continental/smoke" cases in the study, and the cases we showed as "polluted continental/smoke" should be "polluted dust" cases. This mistake has been corrected in the revised version of the manuscript. However, the reviewer points out an important detail regarding the "polluted continental/smoke" and "elevated smoke" classifications. Therefore we have added this information in Section 2.1.

3- I am not entirely convinced by the choice of the three AOD intervals. I believe that using different thresholds for the AOD intervals for different regions and time periods is confusing. For example what you consider 'high' AOD interval for the mixed cases JJA is covering mostly the 'middle' AOD for the smoke cases SO. While a comparable number of profiles for each bin is fairly important for a statistical analysis, I find it more important to compare the same ranges of values. I would suggest either reducing the number of intervals (above a threshold and below a threshold) or choosing values that are applicable to all categories (e.g. in the case of mixed type, have only 2 intervals – as it is clear the AOD values are lower than for smoke).

We understand that the selection of intervals in the way it was done in the manuscript makes the comparison between the two cases (smoke and non-smoke) and the two periods (JJA and SO) hard. For this reason, and according to the reviewer's recommendation, we have decided to remake the figures using fixed intervals of AOD, RH, specific humidity and cloud top altitude (CTA).

4- Is it possible that the different RH profiles between JJA and SO are associated to different meteorological conditions (e.g. wind direction from the ocean, instead from land)? Deaconu et al, 2019 looked at high and low AOD above clouds cases and found that easterlies (winds predominantly from the NE) are associated with larger AODs and

larger humidity values, while the wind coming from the open ocean is characterized by low values of AOD and humidity.

In our analysis, the average moisture (RH and specific humidity) in all three cases (smoke, non-smoke and pristine) is higher during SO than during JJA. We believe that the main cause of the higher moisture values in SO is an increase in the wind speed at aerosol layer altitudes during this period. As shown in Figure 4 in the manuscript, we do not see a clear difference in the average wind direction between JJA and SO for any of the three cases.

The pristine cases (no aerosol above clouds and low RH in the free troposphere) are on average more influenced by winds from the open ocean in both periods (JJA and SO) compared to the aerosol cases (with higher values of RH in the free troposphere). The aerosol cases are more dominated by winds from the NE-E at aerosol layer altitudes. In this way, our results are consistent with Deaconu et al. (2019).

5- The author also mentions that the average monthly horizontal winds show a significant difference between the months of SO compared to JJA at 700hPa. In this paper the authors also separate AOD in different intervals and analyze the RH, but it is not clear if these cases have similar underlying meteorology, and if for all these cases the aerosol and humidity layers are in fact together transported from the land. Considering that the CALIOP Aerosol Profile Data product includes the RH extracted along the track from MERRA-2, but no information on the winds, I suggest mentioning the caveats associated to this approach when comparing these cases and the different resulting heating rates.

We do not see how the elevated aerosol and moisture layers found in the free troposphere could be transported from different directions in the area of study. In particular, since the wind direction is in general clearly related to the presence/absence of both aerosol and moisture. Furthermore, Deaconu et al. (2019) (using a similar area of study) showed that moisture in the free troposphere agrees well with the wind speed

and direction.

We acknowledge that understanding the prevailing meteorology in different AOD intervals is indeed important. By meteorology, we refer here to the atmospheric humidity and wind patterns. In the beginning, we investigated the dominant wind patterns for the smoke, non-smoke and pristine cases, as shown in Fig. 2. It was undoubtedly clear from the wind patterns that the smoke cases had their origins from the nearest biomass burning areas, as expected.

We further observed that as the smoke plume was transported over the open ocean and profiled by CALIOP, we obtained different optical depths in the CALIOP profiles as a function of distance, while having the same wind direction. This makes physical sense since the smoke plume begins to thin out as it travels away from the land-based source. So, in reality, within the same CALIOP transect of the smoke plume, we have profiles that fall into different AOD intervals that we have defined, while having the same wind direction and similar wind speed. Therefore, this relationship between smoke AOD-intervals and wind is not very informative (unless there is a strong reason to believe that the smoke layer can originate from the western side of our study area, which is highly unlikely).

With this in mind, we decided to focus exclusively on investigating the relative humidity as a function of AOD intervals as it potentially has much larger bearing on the AOD itself and the resulting impact on cloud top cooling.

6- The results showing mean profiles of relative humidity for the three aerosol intervals for SO period are surprising. Adebiyi et al., 2015 use radiosondes measurements from St. Helen for different aerosol loading intervals (e.g. Fig 11, Fig 14) and their results show both the humidity and temperature boundary layer-top inversions strengthen as the aerosol loading above increases. Flight campaign measurements that took place in the Southeast Atlantic during the biomass burning season (ORACLES, CLARIFY and AEROCLO-sA) have also found increased moisture with increased aerosol loading.

Would the authors like to comment on this?

We have added a section (Section 3.6) with a deeper analysis related to these results. We observe that during JJA, the average moisture in the free troposphere (between 2 and 7 km) and the temperature between 1.5 and 2 km (approximately an average of boundary layer top inversion altitudes) increase with the average AOD in agreement with previous studies. During SO, the relation between temperature and AOD is similar as in JJA, while the relation between AOD and moisture is weak. Specifically, we find no correlation between moisture and AOD in September. We believe that several factors could cause the difference between our results and previous studies. For instance, our study area is different compared to Adebiyi et al (2015) and thus the meteorology may be different. It is also possible that we have differences in the hygroscopicity of the aerosols in the different areas. In Adebiyi et al (2015), the study region is further from the continent compared to our work, i.e. aerosols that originally consisted of soot may have had time to become aged and more hygroscopic; this may affect the AOD-atmospheric moisture relation. We also used different observational data and we did not analyse exactly the same period as Adebiyi et al. (2015) and Deaconu et al. (2019). We can also have differences due to sampling. For instance, Adebiyi et al. (2015) looked at averages over a large number of profiles during 2000-2011, which are shown in their Fig 11. But the same figure also shows a large variability of moisture in the free troposphere for each of their aerosol optical depth terciles. In addition, we do not discard the possibility of having an additional source of errors due to e.g. AOD inaccuracies in CALIPSO V4 or because RH values are not measured by CALIPSO but taken from MERRA-2.

7- Fig.6: Could the authors also plot specific humidity profiles? Even in a supplementary Figure. The authors mention 'no indication of a semi-direct effect'. However, there is no additional study on the cloud liquid water path and cloud optical thickness that would support this comment. Their statement is based on the results on the LW cooling rates due to RH, which, as they mentioned, shows confounding impacts due to the

variability of the cloud top cooling rates. This study would be enhanced with a sensitivity test, in which the cloud top altitude would be fixed for the different aerosol types, AOD and RH intervals and time periods.

The profiles of specific humidity are now shown in all figures where profiles of RH are also shown (e.g. Figure 6 ).

We have reformulated the statement regarding the 'semi-direct aerosol effect'. What we meant is that our specific data and analysis did not show any indication of such an effect. However, we do not exclude that a semi-direct effect actually exists.

Reply to referee #2 comments on manuscript:

The study examines the impacts of smoke and other aerosols species on the low-level clouds over the southeast Atlantic. The paper's main goal is to characterize the effects of aerosol loading, aerosol types, and the mid-tropospheric relative humidity on the thermodynamical profile, radiative heating rates, and the cloud-top radiative cooling. To do so, the study used aerosol and cloud retrievals from CALIPSO and CloudSat as well as MERRA-2-based meteorological variables obtained as part of CALIPSO and ERA5 reanalysis dataset (see Table 1). The study focused on the area between 10 to 18S and 2 to 10E, and between June and October 2007 and 2010. According to the study, the justification of the area and study period is because the low-level cloud and above-cloud aerosol maximize sometimes between June and October. In addition, the authors also added that the months of study were chosen in other to compare their results with previous studies of (Deaconu et al., 2019) and (Adebiyi et al., 2015), who used June-August and July-October period, respectively. Furthermore, the study divided their dataset into three cases: (1) the smoke cases – based on the aerosol retrieval classified as "elevated smoke" in CALIPSO. These are cases where the cloud layers (between 0.75 km and 2.5 km) are separated from the "elevated smoke" layers by at least 0.4 to 6 km. (2) The mixed cases are the same as the smoke cases, except that they are for other CALIPSO aerosol retrievals NOT classified as "elevated smoke".

(3) The pristine cases with cloud layers between 0.75km and 2.5km but no aerosol above. The study finds that the smoke and mixed cases have similar meteorological conditions (winds direction, thermodynamical profile), although with different magnitudes. The study also suggests that no monotonous increase in the mid-tropospheric relative humidity with increasing aerosol optical depth exists. In addition, they find that while the vertical distribution of the SW heating rates corresponds to the above-cloud aerosol extinction for both the smoke and mixed cases, they do not correlate with the moisture distribution. Given the importance of the southeast Atlantic's aerosol-cloud interaction to the global radiative budget, this article will add to the growing body of knowledge about the complicated aerosol-cloud-meteorology system over the southeast Atlantic. While the article is generally well written, I have some comments that I hope the authors will address.

We thank the reviewer for the encouraging remarks and constructive comments.

1. The authors should change the title. While smoke is undoubtedly an absorbing aerosol, I don't think the authors have sufficiently argued that the "mixed cases" are indeed "non-absorbing" aerosols. To my understanding, these mixed cases also contain smoke and dust, which are both absorbing aerosols (Samset et al., 2018).

This is a good recommendation and we agree with the referee; therefore we have changed the title to: "Impact of smoke and non-smoke aerosols on radiation and low-level clouds over the Southeast Atlantic from co-located satellite observations."

2. That said, I think calling other CALIPSO-derived aerosol types that do not fall within the "elevated smoke" classification "mixed cases" might be somewhat inaccurate. The reason is that the term "mixed" gives the impression there are always more than one aerosol species within the aerosols layer at all times, different from the elevated smoke. Although Fig. 1 shows cases of polluted/continental smoke, dust, and dusty marine aerosols, it is unclear whether those cases represent the entire atmospheric column or only the mid-troposphere. Moreover, the lidar ratio used to separate the elevated

smoke in CALIPSO V4 is identical to that of polluted smoke, except that elevated smoke is defined as smoke in the mid-troposphere. Hence, if indeed the authors focused on aerosols above clouds, then part of the polluted smoke included with the mixed cases is in-fact misclassified elevated smoke in the mid-troposphere, which is expected to have similar optical properties, and therefore similar impacts on meteorology and low-level clouds as the elevated smoke. Hence, in addition to changing the classification name, the authors need to make a stronger case that the optical properties of the "mixed cases" are in fact, different from those of "smoke cases". The author should also show that a significant amount of dust and smoke are represented at all height levels of the mid-troposphere to justify their "mixed cases" argument. Perhaps Fig. 3 can be done for the different aerosol species identified in Fig. 1.

We have decided to rename the "mixed" cases as "non-smoke" cases in response to the referee's comment. Regarding the concern about the aerosol cases representing the atmospheric column or only the mid-troposphere: all aerosol cases shown in Fig.1 are aerosols above clouds and we show the altitudes at which the aerosol layers were found in Fig. 3 (together with the cloud top altitudes). We have clarified in the text (when referring to Fig. 1) that these aerosol layers are above clouds. We have also clarified in section 2.1 that the "non-smoke" aerosol cases correspond to aerosol above clouds.

There was a mistake with respect to the aerosol classification in Fig.1 in the previous version of the manuscript. There are no "polluted continental/smoke" cases in the study, and the cases we showed as "polluted continental/smoke" should be "polluted dust" cases. This mistake has been corrected in the revised version of the manuscript. After making these changes, we believe that the rest of the referee comments are also addressed.

3. The author should discuss the differences between this study and that of Adebiyi et al. (2015) and Deaconu et al. (2019), given that the conclusions in this study are sufficiently different from the other two studies (to some degree). Although some comparisons were made in some parts of the texts, it is mostly incoherent and unclear to the reader looking to know how exactly this study stands out from the previous papers. I will suggest that the authors have a separate sub-section, where they discuss specifically the differences between this study and previous ones.

Following the reviewer's suggestion, we have added Section 3.6 where differences between the results in our study and those of Adebiyi et al. (2015) and Deaconu et al. (2019) are discussed.

4. The authors did not explore the caveat that CALIPSO bottom layers have substantial uncertainty, as have been shown in previous studies. For example, CALIPSO has been shown to have a significant bias when compared to aircraft-based observation from HSRL (e.g. Kacenelenbogen et al., 2011), and satellite-based observation from CATS (e.g. Rajapakshe et al., 2017). The authors should account for this uncertainty in their analysis. In addition, the authors should also discuss how this caveat could affect their conclusions.

Based on the reviewer comment and the analysis in Rajapakshe et al. (2017), we have realized that 0.4 km is probably a too small distance to ensure that aerosols and cloud layers are detached. For this reason, we have decided to increase the distance to 0.75 km, which is the same value used by Constantino and Breon (2013) in their "well separated cases" (aerosol layers separated from cloud layers). After increasing the distance, the number of "aerosol above cloud" cases decreased and all the figures of the manuscript have been remade with this reduced dataset.

5. Surprisingly, the authors find decreasing RH as a function of AOD. First, as indicated below, I think the same AOD and RH intervals should be used for all cases highlighted in Fig.6-8 ). That puts the results on equal footing and allows for better comparison. Secondly, I wonder if something is inherently mistaken in the dataset that the authors used, since it is wildly different from the previous study. Adebiyi et al. 2015 used radiosonde collected at St. Helena Island and found a reasonable relationship

between AOD and moisture in the mid-troposphere. Other studies that used reanalysis datasets have found a similar positive relationship. Here, the authors used MERRA temperature and relative humidity values packaged with CALIPSO retrievals, not those directly obtained from the MERRA website. While I do not suggest that these datasets may be different, I think the author should validate their results using other reanalysis datasets. Specifically, the authors should use the ERA (and possibly NCEP) reanalysis and show that the same result holds as those found in this study. Those results/plots should be included in the supplementary document and cited in the paper. It is worth noting that these reanalysis datasets are often significantly different, especially over the ocean where limited ground-based constraints are assimilated. Nevertheless, as Adebiyi et al. 2015 (see their Fig. 14) showed, the reanalysis datasets are expected to be broadly consistent. Third, I wonder if other criteria set in the analysis also do play a role in the differences? The authors could use their datasets and test their results over the same regions and periods defined in other studies (Adebiyi et al., 2015; Deaconu et al., 2019). Finally, given that this part of the study is in sharp contrast from other studies, the authors must spend a sufficient amount of time explaining to the reader that these results are accurate, and also discuss in detail the difference between this study and others that found different results (see comment above).

We have double checked our results and added a section (Section 3.6) with a deeper analysis related to these results. According to the reviewer suggestion, we now also use the same AOD and RH intervals for all cases. We observe that during JJA, the average moisture in the free troposphere (between 2 and 7 km) and the temperature between 1.5 and 2 km (approximately an average of boundary layer top inversion altitudes) increase with the average AOD in agreement with previous studies. During SO, the relation between temperature and AOD is similar to JJA, however, the relation between AOD and moisture is weak. Specifically, we find no correlation between moisture and AOD in September. We believe that several factors could cause the differences between our results and previous studies. For instance, our study area is different compared to Adebiyi et al (2015) and thus the meteorology may be different.

It is also possible that we have differences in the hygroscopicity of the aerosols in the different areas. In Adebiyi et al (2015), the study region is further from the continent compared to our work, i.e. aerosols that originally consisted of soot may have had time to become aged and more hygroscopic; this may affect the AOD-atmospheric moisture relation. We also used different observational data and we did not analyse exactly the same period as Adebiyi et al. (2015) and Deaconu et al. (2019). We can also have differences due to sampling. For instance, Adebiyi et al. (2015) looked at averages over a large number of profiles during 2000-2011, which are shown in their Fig 11. But the same figure also shows a large variability of moisture in the free troposphere for each of their aerosol optical depth terciles. In addition, we do not discard the possibility of having an additional source of error due to e.g. AOD inaccuracies in CALIPSO V4 or because RH values are not measured by CALIPSO but taken from MERRA-2.

6. Lastly, the details of all datasets should be discussed in the methodology. In particular, a lot of this study's results rely on the CloudSat-derived radiative fluxes. While those radiative fluxes may have used CALIPSO aerosol extinction profiles, they make other important assumptions about the aerosol's optical properties that can have important implications on this paper's conclusion. The authors must discuss the limitation of those assumptions in their results. In addition, there is a potential difference in the thermodynamical profiles used to obtain those CloudSat-derived fluxes (ERA) and those used for general characterization of meteorology in this study. As suggested above, the author must show that both reanalysis datasets support their overall conclusion.

The description of the 2B-FLXHR-LIDAR product in section 2.1 has been extended. It is true that there is a potential difference between the thermodynamical profiles used to obtain the CloudSat-derived fluxes (ERA) and those used for the general characterization of meteorology (specifically RH and potential temperature) in this study (MERRA-2). For these reasons we have now decided to use the CloudSat ECMWF-aux product (instead of MERRA-2) for the characterization of meteorology in the study, as it contains ancillary ECMWF variables (including temperature, pressure and specific humidity) that are involved in the computation of CloudSat-derived fluxes. However, in the new Section 3.6 we use the atmospheric variables from MERRA-2 added to CALIPSO.

We have compared RH, specific humidity and potential temperature from the CALIPSO Aerosol Profile Data product (MERRA-2) and the CloudSat ECMWF-aux product (ECMWF) and they show a good general agreement (see Figure 1 in supplement).

Other Comments 1. Section 2.1: Given that the study relies on elevated smoke versus other pollution, some more details should be provided on how CALIPSO made those classifications.

Additional information has been added to Section 2.1.

2. Line 169: I am finding it difficult to reconcile why the number of profiles shown in Table 2, for smoke cases, for example, is higher than the total number of cases of elevated smoke shown in Fig. 1. I understand that the "smoke cases" should be a subset of the CALIPSO's "elevated smoke" classification since it has some additional criteria. The same discrepancy appears to be present in mixed cases as well. Are they strictly cases where the aerosol layer is above the low-level cloud for both the smoke and mixed cases? That is what Section 2.3 appears to suggest. The difference between Fig. 1 and Table 2 should be further clarified.

There was a mistake with the numbers in Table 2, they should be in agreement with Figure 1. We thank the reviewer for pointing this out and have now corrected the mistake. In Table 2, the number of profiles of smoke cases correspond to the elevated smoke shown in Figure 1. For the non-smoke cases the number of profiles is the sum of the remaining aerosol types in Figure 1. This is clarified in the caption of Table 2. Furthermore, the caption in Figure 1 was slightly modified so that it is easier for the reader to make a connection between Figure 1 and Table 2.

3. Line 175-178: Based on previous studies, like some of those cited by the authors, I think it is unlikely to have more cases of aerosols above clouds between 2-6E than

6-10E. I will suggest that the authors make the same plot as Fig. 3 but separated as a function of longitude or latitude. Such a figure can be included in the supplementary.

We think there may be a misunderstanding. The smoke, non-smoke and pristine distributions shown in Figure 2 are strongly influenced by our selection criteria, in particular that we require a certain distance between the aerosol layer and the cloud. Therefore, our results do not necessarily imply that situations with aerosols above clouds are more frequent between 2-6E compared to 6-10E. In fact, Figures 3 and 4 of Devasthale and Thomas (2011) show that we can have a similar number of cases from 2E to 10E. Please note that Figure 2 has been updated due to a new threshold criterium (0.75 km instead of 0.4 km) in the separation between aerosols and clouds as well as due to a previous mistake in the data processing for the pristine cases between 2 and 3E.

4. Figure: While it may already be mentioned in the text, Fig. 2 should include the latitude range considered for the longitude distribution. Similar thing for the latitude distribution. Also, for Fig.3.

The latitude (longitude) ranges for the longitudinal (latitudinal) distributions were added to both figures.

5. Line 208: Change "... southerly component compared to the two aerosol cases" to "... southerly winds compared to the two aerosol cases."

The paragraph has been reformulated and the sentence has been removed.

6. Line 227: Please re-write the sentence.

The sentence was re-written.

7. Line 214 should be Figure 5a-c.

Thanks for pointing this out. It is now Figure 5a-d due to the addition of the specific humidity profile in Fig 5c.

8. Line 231 should be Figure 5d–?

It is now Figure 5e-f due to the addition of the specific humidity profile in Fig 5c.

9. Section 2.1 There should be a brief description of how the radiative heating rate of Cloudsat is obtained to give the reader the complete picture and the ingredients to interpret the result adequately. For example, while CloudSat may have used the vertical extinction profile from CALIPSO, what spectral distribution of single scattering albedo and asymmetric factor are assumed? What surface parameters that could affect radiative heating is assumed? How do they treat other wavelengths that are not provided by CALIPSO in longwave and shortwave? The author needs to provide a complete context to interpret the result.

The information has been added to section 2.1.

10. Fig. 5 (as on others, see comments above) – always provide complete information for each figure. In this case, information of what data is plotted should be mentioned in the comment section.

We have updated the information and it should now be complete.

11. Fig 6: Again, information needs to be complete. Where is the AOD coming from?

AOD refers to the Column Optical Depth Tropospheric Aerosols at 532 nm from the CALIPSO Aerosol Profile Data Product. This has been clarified in the caption to Fig.6 and explained in the description of Table 1.

12. Like 249-254: The two paragraphs are repeated. The author should carefully go through the manuscript before any future re-submission.

Thank you for pointing this out. The text has been corrected.

13. Line 256-58: Any comparison between different cases will likely not be "fair" – that is, it will result in comparing "oranges" against "apples". I will suggest that the authors make the tercile range the same for the AOD classes.

We have remade the figures using fixed intervals of AOD for all cases.

14. Section 3.5: If AOD will be used to make inference about RH with the aerosol plume, it must be AOD of the aerosols above the cloud. It is difficult to expect a clear linear relationship between RH averaged within a layer and column integrated AOD.

We believe this is a misunderstanding. The AOD values correspond only to the aerosol(s) layer(s) detected above clouds. The column integrated AOD (Column Optical Depth Tropospheric Aerosols at 532nm) corresponds to the same profiles of aerosol extinction that are shown in the different figures containing this variable and to the distribution of aerosol layers shown in Fig 3. There are no aerosol layers and there is no aerosol extinction outside that range of altitudes observed in our cases (aerosols are always above cloud) since we filter out other situations.

15. Line 275: Is this RH averaged within the aerosol layer or the entire atmospheric column? Please clarify this within the text and in the figure caption.

It is the average RH between cloud top and 7km. This is now clarified in the text and caption.

16. Fig A1: Check the caption. AOD or RH?

RH is the correct variable. The mistake has been corrected.

Please also note the supplement to this comment:
https://acp.copernicus.org/preprints/acp-2020-1089/acp-2020-1089-AC1-supplement.pdf
* * *
[Figure]

**Supplement:**

[Figure]

Figure 1: (a-d) Average aerosol extinction, relative humidity ($RH$), specific humidity ($q_v$) and potential temperature ($\theta$) profiles for the smoke, non-smoke and pristine cases during JJA and SO using the values extracted from CALIPSO Aerosol Profile data product (MERRA-2). (e-f) Same variables, but extracted from the ECMWF-aux files from CloudSat, except for the aerosol extinction which comes from CALIPSO.